# Influence of Oxidation Processing Temperature on the Structure, Mechanical and Tribological Properties of Titanium Using Carbon Sheets

Tong Chen [1],*, Shinji Koyama [1], Shinichi Nishida [1] and Lihua Yu [2]

1 Mechanical Science and Technology, Graduate School of Science and Technology, Gunma University, Gunma 371-8510, Japan; koyama@gunma-u.ac.jp (S.K.); snishida@gunma-u.ac.jp (S.N.)
2 School of Material Science and Engineering, Jiangsu University of Science and Technology, Zhenjiang 212003, China; lhyu6@just.edu.cn
* Correspondence: t182b602@gunma-u.ac.jp; Tel.: +81-277-30-1659

**Abstract:** Surface processing of pure titanium was performed using a carbon sheet to increase the surface hardness and improve tribological property. The effect of processing temperature (750–950 °C) for 2 h on the structure, mechanical and room-temperature tribological properties of the treated samples was investigated using X-ray diffraction, scanning electron microscopy, and ball-on-disk tribometry, respectively. The Gibbs free energy was also calculated to evaluate the compounds generated at different processing temperatures. As a result of the examination, the hardened layer was mainly composed of titanium oxide and titanium carbide. With the increasing processing temperatures, the thickness of the hardened layer increased first and then decreased gradually. It was also revealed that the surface hardness was increased first and then decreased as the processing temperature increased. The fricative value of the treated samples showed a minimum value of 84.1 dB for a processing temperature of 850 °C. The depth and width of the wear tracks increased first and then decreased gradually with the increasing processing temperatures. The worn surface of the treated samples at higher temperatures showed a very good wear resistance. A processing temperature at 850 °C is considered optimal as it provides sufficiently high hardness and a low coefficient of friction to reduce fricative during practical use.

**Keywords:** processing temperature; titanium; structure; mechanical property; tribological property





## 1. Introduction

Titanium and its alloys, exhibiting special mechanical properties and excellent corrosion resistance, are widely used in personal digital assistants, heart stents, shell materials of electronic devices, and so on [1]. In the widespread applications of the materials, the surface of titanium shows relatively poor mechanical properties, and the improvement of the surface layer is strongly demanded for longer service-life of the various apparatus. To solve this problem, a series of industrial techniques have been developed [2], including deposition and diffusion [3–6]. The main purpose of surface treatment is to improve the abrasive resistance of titanium, extend its service-life, and reduce the waste of resources.

Studies have shown that surface treatment technology can improve the wear resistance, corrosion resistance, biocompatibility and other properties of titanium and titanium alloys, thereby further expanding its application range. Commonly used surface modification methods are thermal oxidation, micro-arc oxidation, physical vapor deposition and laser cladding. Among them, thermal oxidation treatment is a simple, economical and effective surface modification method, which can greatly improve the friction and wear resistance and corrosion resistance of titanium and titanium alloys [7–9].

Through thermal oxidation surface treatment technology, a thicker hard oxide film composed of the titanium oxide layer and oxygen diffusion layer can be formed on the

surface of titanium and titanium alloy. The oxide film can effectively protect the material and improve its wear resistance.

Bansal et al. investigated the influence of the oxidation temperature to improve the corrosion resistance of pure titanium [10]. Maytorena-Sánchez et al. discussed how the formation of a $TiO_2$ coating can improve the hardness by oxidation [11]. Du et al. reported the high-temperature corrosion of Ti and Ti−6Al−4V [12]. They found that adding Al into the simple boronized coating is beneficial for the high-temperature oxidation resistance. Most current articles on oxidation are about the oxidation of alloys. Articles about the oxidation of pure titanium are limited to single studies of its corrosion or mechanical properties, and there are few studies on the overall structure, mechanical, and tribological properties of pure titanium. In addition, based on Reference [12], this paper proposes a conjecture about whether the use of carbon cloth will improve the performance of pure titanium during oxidation treatment, and conducts specific studies on the structure, mechanics, and friction and wear properties. Many researchers have research pure titanium preliminarily, but there are few reports about oxidation by using carbon sheet.

This study aimed to synthesize a continuous thick hardened layer on pure grade-2 Ti by processing in a furnace using the carbon sheet. The effects of the heat-treatment condition (processing temperature) on the structure, and mechanical and tribological properties of the hardened layer were investigated. To understand the development of the hardened layer, the Gibbs free energy of possible compound reactions with titanium was calculated using the National Institute of Standards and Technology Janaf (NIST-JANAF) thermochemical table. The hardened layer formed on the pure Ti surface provided a protective coating against serious wear loss during the service period. Therefore, the correlations between the processing temperature and properties, including the fricative value, are discussed in this research report.

## 2. Experimental

The samples used in this study comprised a $50 \times 20 \times 5$ mm$^3$ plate cut from the as-received plate using fine cut machining (HS-45A2, HEIWA TECHNICA, Kanagawa, Japan). Pure grade-2 Ti (composition in wt.%: 0.2 O, 0.003 N, 0.013 H, 0.25 Fe, 0.008 C, and remainder Ti) was used as the substrate material. The surface to be treated of the Ti was finished by grinding with emery paper (grade 4000). The pure titanium is heated by the air furnace in the atmosphere by using the carbon sheet. The schematic diagram of the sample setup (a) and the experimental apparatus of the wear and fricative testing (b) were shown in Figure 1. In addition, a scanning electron microscope (SEM, S-3000N HITACHI, Tokyo, Japan) image of the carbon sheet (c) was also shown in Figure 1. The elemental distribution of the cross-sectional samples was determined by energy dispersive spectroscopy on an energy dispersive analyzer (EDX, SEDX-500, Shimadzu, Tokyo, Japan). The titanium plate was placed between two pieces of carbon sheet and the load at 1 MPa was applied to it. Processing temperatures were changed from 750 to 950 °C for 2 h in the atmosphere. To determine the compound formed on the surface after processing, the treated surfaces were investigated using an X-ray diffractometer (XRD, 2200VF, Rigaku, Gunma, Japan) with CuKα radiation working at the optimum voltage of 32 kV and anodic current of 20 mA. Furthermore, the cross-section of treated samples was prepared by ion milling (IM4000, HITACHI Tokyo, Japan) with 7200 s. Microstructural and morphological characteristics of hardened layers were examined using a SEM. Surface hardness testing was carried out throughout the hardened layers of the samples using a Vicker hardness tester (HMV-1, SHIMADZU, Kyoto, Japan) with the applied load is 0.98 N to gain the hardness of the hardened layer more accurately and at a dwell time of 15 s. In order to reduce the measurement error, the sample was measured for 5 times and averaged. Ball-on-disk dry sliding tests were performed at room temperature to evaluate the tribological properties of the samples using a tribometer (FPR-2000, Rhesca, Tokyo, Japan) with a zirconium dioxide ($ZrO_2$) ball and a counter-face with a radius of 2380 μm. Samples with the dimensions of $50 \times 20 \times 5$ mm$^3$ were used for wear testing with a sliding linear speed

of 200 mm/s, an applied force (loading force during the wear testing) of 4.9 N, and a test time of 3600 s (corresponding to a sliding distance of 720 m). All tests were performed at room temperature. The microstructural and morphological features of the wear track after wear testing were examined using SEM under secondary electron imaging mode at a voltage of 15 kV. The fricative value is measured by a condenser microphone (TM-103, TENMARS, Tokyo, Japan) installed 20 mm above the contact point between the ball and sample. The depth profile of the sliding portion after the wear testing was measured by a laser microscope. In order to reduce the measurement error, the sample was measured four times and averaged.

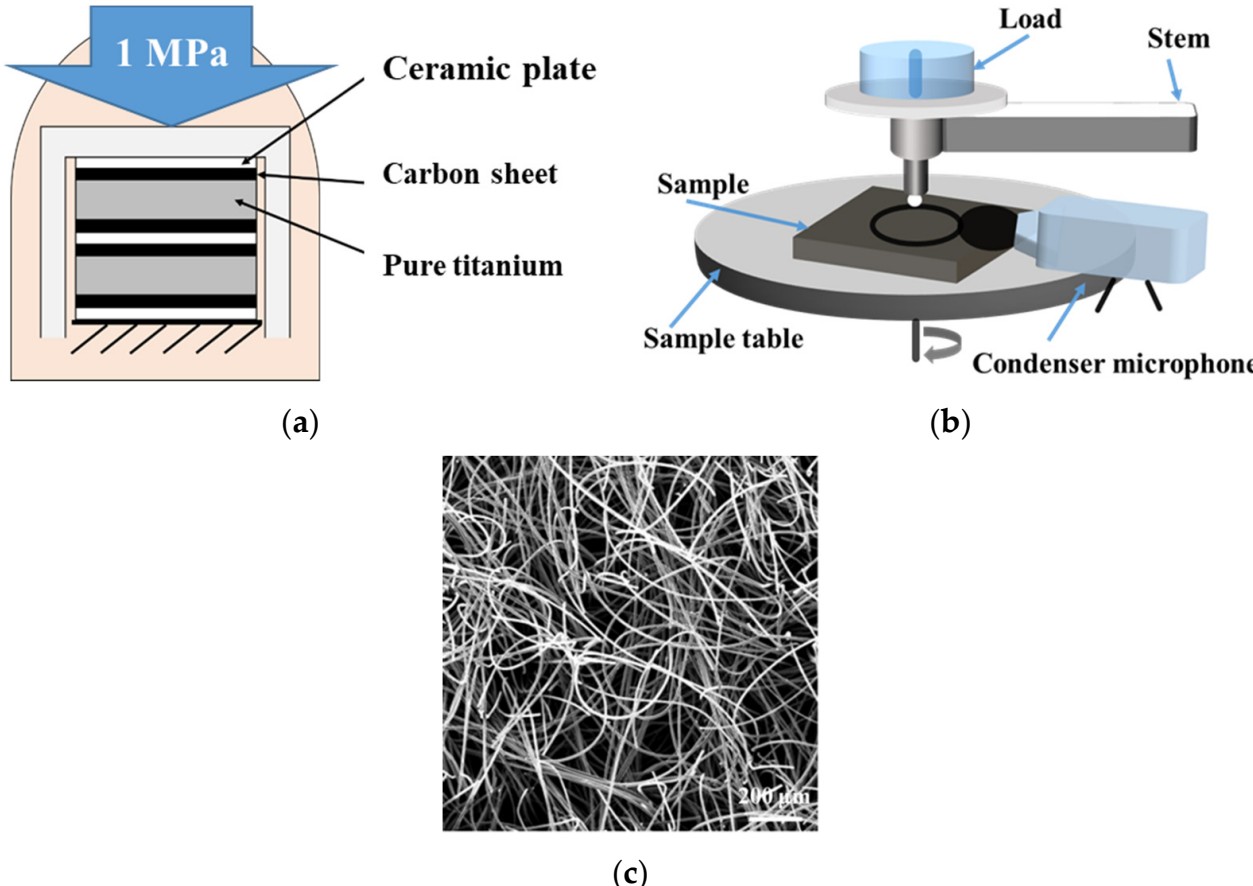

**Figure 1.** Details of the experimental work (**a**) schematic diagram of the sample setup, (**b**) experimental apparatus of the wear and fricative testing and (**c**) SEM carbon sheet image.

## 3. Results and Discussion

### 3.1. Structure Analysis

Figure 2 shows the X-ray diffraction (XRD) patterns of the surface of samples with different processing temperatures. From the XRD results, it can be seen that as the processing temperature increased, the oxide/carbide layer on the surface of the sample is gradually thickened. The intensity of the diffraction peak of the substrate phase gradually decreases. As shown in Figure 2, it was revealed that $TiO_2$, $Ti_2C$, and $Ti_8C_5$ were formed on the surface. It was inferred that oxygen diffused at high temperatures and became more pronounced with different processing temperature. Rosa. et al., discussed the oxygen diffusion in alpha and beta titanium in the temperature range of 932 to 1142 °C [13]. It was found that the diffusion rate of oxygen in beta titanium is faster than that in the alpha titanium for the same processing time. For beta titanium, the diffusion rate becomes slower as the processing temperature increases. Under the same test conditions, the X-ray detection depth is constant, and the thicker the hardened layer, the lower the detection sensitivity

of the substrate peak, resulting in a decrease in the intensity of the titanium peak. When the processing temperature at 950 °C, the intensity of the oxides diffraction peak is the highest, which indicates that the content of oxides formed on the surface is the highest at this processing temperature.

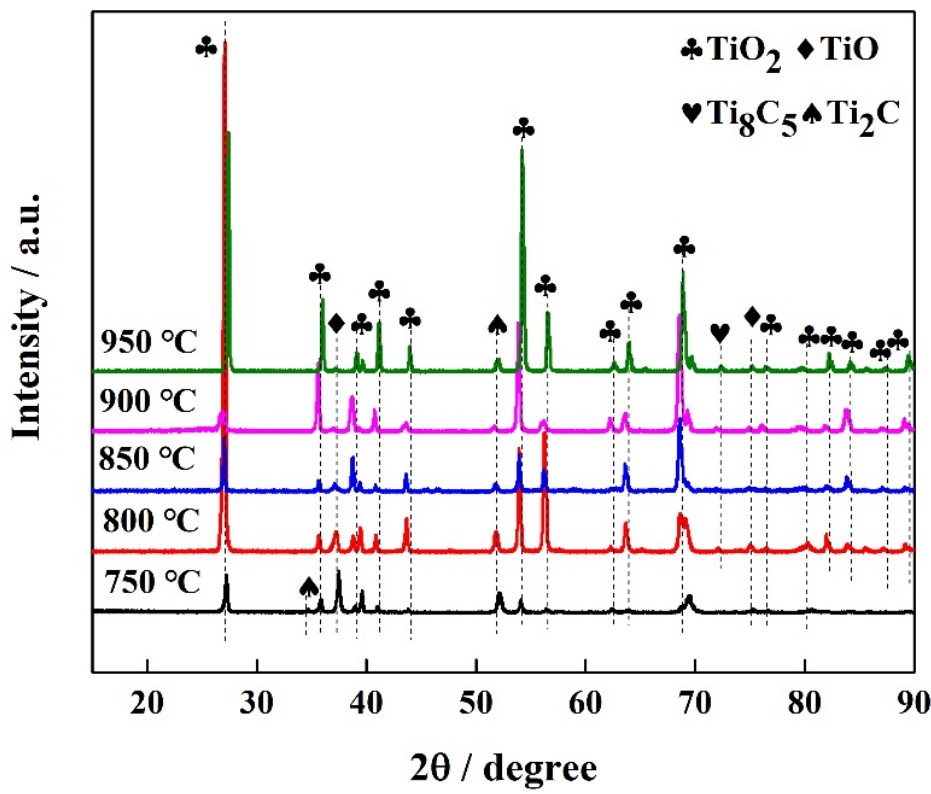

**Figure 2.** X-ray diffraction patterns of the surface of samples with different processing temperatures.

Figure 3 shows the ΔrG profile of the compound reactions in the temperature range from 0 to 2250 °C, as calculated using the NIST-JANAF thermochemical numeric data [14]; and the possible reactions are shown below:

$$C + O_2 = CO_2 \tag{1}$$

$$CO_2 + C = 2CO \tag{2}$$

$$2C + O_2 = 2CO \tag{3}$$

$$2CO + O_2 = 2CO_2 \tag{4}$$

$$2Ti + 2CO = 2TiC + O_2 \tag{5}$$

$$Ti + O_2 = TiO_2 \tag{6}$$

$$TiO_2 + 3C = TiC + 2CO \tag{7}$$

The influence of processing temperature on the Gibbs free energy (ΔrG) of Reactions (1)–(7) are shown in Figure 3. As is well known, the reaction can only occur when the ΔrG is less than zero, and the smaller the ΔrG, the easier the reaction occurs [15]. When carbon and oxygen react, the reaction speed is controlled by the amount of air convection. In the case of sufficient oxygen, carbon reacts with oxygen in the air to produce carbon dioxide, and then carbon dioxide reacts with carbon to produce carbon monoxide. The equations are shown in (1) and (2). The other is that when the oxygen is insufficient, carbon monoxide (3) is generated by the reaction between carbon and oxygen in the air, and then the carbon monoxide further reacts with oxygen in the air to form carbon dioxide (4), and then the

carbon dioxide reacts with the titanium. The carbon reacts to produce carbon monoxide, and the reaction equation is the same as (2). It can be seen from Figure 3 that the reactions of (1), (3), (4), and (6) can occur at this study conditions. In this paper, the weight loss analysis of the carbon fiber treated at the processing temperature of 850 °C for two hours was researched. The results are shown in Table 1. In the absence of external wear, only the carbon sheet was subjected to heat treatment experiments in the atmosphere. After the processing, the weight of the carbon sheet has changed, which can be considered as an oxidation reaction of the carbon sheet with a certain substance in the atmosphere. Based on this results, it found that under the same processing time, the weight of carbon fiber is reduced. This can indicate that the carbon atoms in the carbon fiber have reacted with oxygen in the air, causing the mass to decrease. As shown in the reactions of (1) and (3). References [16] and [17] also show that reactions (1) and (3) can occur, generating carbon dioxide and carbon monoxide, respectively. This confirms the hypothesis that carbon atoms react with oxygen atoms in the air. From the results of formulas (5) and (7), it can be seen that the $\Delta rG$ of (5) is high than zero, which means that the reaction cannot proceed. In addition, it also can be seen that (7) can occur when the temperature is high enough. The processing temperature in this research did not reach the temperature at which the reaction can occur. Therefore, in this paper, TiC cannot be formed by the reaction of (7). It also can be known from the titanium-carbon phase diagram that only when the carbon content is high, TiC can be formed.

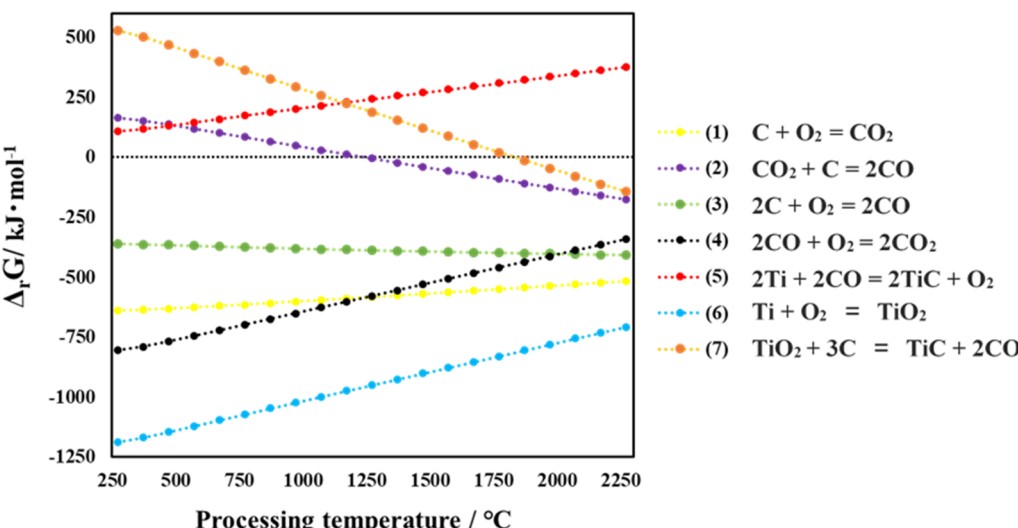

**Figure 3.** Relationships of processing temperature on the Gibbs free energy $\Delta rG$ of reactions (1)–(7).

**Table 1.** Weight loss testing of the carbon fiber at 850 °C.

| Processing Temperature/°C | Before | After |
|:---:|:---:|:---:|
| 850 | 0.077 g | 0.051 g |

Furthermore, oxidation is a process in which oxygen atoms diffuse from the outside to the inside and react with titanium. With increasing the processing time, the content of oxygen on the surface and inside of the sample is different. This result is also mentioned in [18]. Abuluwefa et al. also studied the diffusion of oxygen with different temperatures. It shows that as the processing time increases, the oxygen content inside the sample increases and reaches a compound that maintains a stoichiometric ratio [18]. The amount of oxygen on the surface of the sample is large, which plays a leading role in the reaction with titanium. At this time, Ti + $O_2$ = $TiO_2$ (6) reaction is likely to occur, and $TiO_2$ is formed on the outermost surface. This is also the reason why a large amount of $TiO_2$ was detected at higher temperature.

To determine the effect of the processing temperature on the cross-sectional metallic structure of a hardened layer, the SEM observation, and EDX analysis were carried out. As shown in Figure 4, the hardened layer varied with different processing temperature were observed in the vicinity of the surface. In each processing temperature, there was a layered structure with dark contrast different from the base metal near the sample surface. Besides, the thickness of this dark contrast layered structure tended to increase with increasing processing temperature. This shows that at each processing temperature, a new substance is formed on the surface of the sample. These new substances appear dark contrast layered under the SEM. King et al. also showed that during the oxidation process, a layer of dark layered substance was formed on the surface of the sample under the SEM [19]. With the increasing processing temperatures, the thickness of the dark contrast layer increased first and then decreased gradually. From the EDX analysis results, it was found that the C and O detection areas coincided with the layered structure that had a dark contrast different from the substrate near the sample surface. Therefore, it is considered that oxides were generated in the dark contrast region by oxygen diffusion and infiltration treatment.

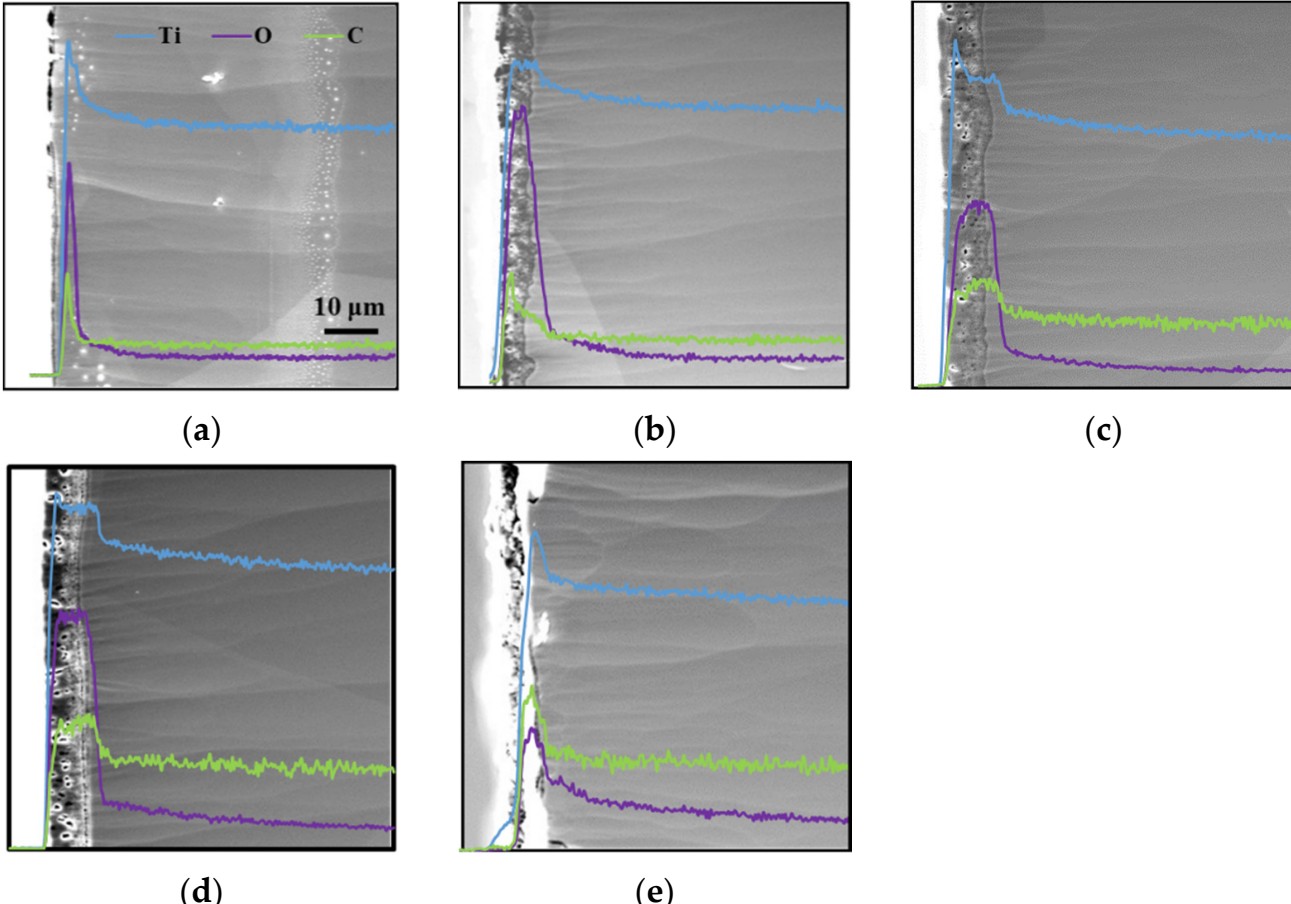

**Figure 4.** SEM micrographs and EDX analysis of the cross-section with different processing temperatures: (**a**–**e**) 750, 800, 850, 900, and 950 °C.

From Figure 4e, it was found that the hardened layer peels off and the content of C and O decreased significantly at this temperature. From these results, it was due to high-temperature embrittlement, the compounds formed on the titanium surface were prone to peel at 950 °C. Chan, et al. studied the dynamic embrittlement and oxidation-induced cracking in superalloys. It found that oxidation-induced crack growth plays a leading role in metal embrittlement [20]. In addition, the existence of residual stress also leads to the hardened layer peel off, show the following discussion for details. Transformation of α-Ti

⇆ β-Ti is known to occur around 880 °C, α-Ti has a dense hexagonal structure, and β-Ti has a body-centered cubic structure is there. At processing temperatures of 750 and 800 °C, O and C were strongly detected on the surface of the test piece and tended to decrease gradually. On the other hand, when the processing temperatures is 800 and 850 °C, O and C were strongly detected from the sample surface to the entire layered structure. This suggests that the diffusion of O and C in β-Ti was faster than the diffusion of O and C in α-Ti. This result is also mentioned in [13,21]. Besides, it is considered that cracking occurred when cooling from 850 °C or higher due to lattice transformation.

From the above results, it was found that at processing temperatures of 750, 800, 850, and 900 °C, diffusion of O and C was promoted, and a compound layer was formed near the surface. However, when the processing temperature is 950 °C, in order to alleviate the high compression stress of oxygen and carbon supersaturated solid solution, it is thought that exfoliation of the compound layer occurred. This will lead to the hardness of the hardened layer decreasing.

### 3.2. Mechanical Property

Figure 5 shows the surface hardness (H) of untreated and treated samples with applied load at 0.98 N. From Figure 6, the H was increased first and then decreased as the processing temperature increased. The H of the untreated sample was about 258 HV, whereas the hardness reached the maximum value was 1286 HV at 850 °C. As shown in the XRD results in Figure 2, titanium oxide and titanium carbide were detected strongly on the surface of the samples as the processing temperature increased. Sivakumar Bose et al. revealed that the hardness of $TiO_2$ is about 800 HV, and the hardness of TiC is about 3200 HV. This may be due to the presence of $TiO_2$ and TiC causing the hardness to increase [22]. However, the hardness value measured in present research is less than the hardness value in the Reference [20]. It was due to the diffusion layer was thin, and then the base material was affected when measuring the H. Besides, the ratio of $TiO_2$ in the product is large, and the influence on the hardness of the sample is larger than TiC. This is also the reason why the hardness has not increased significantly. However, when the processing temperature increased to 950 °C, the H was greatly decreased due to peeling off of the hardened layer. This was consistent with the SEM observation results. In addition, when the processing temperature is higher than the phase transition temperature, as the processing temperature increases, the movement of molecules becomes more violent, and the diffusion speed of oxygen in the β phase slows down, making the space between titanium atoms and oxygen atoms increase [13]. When the temperature decreases to room temperature, the performance of the sample at this processing temperature is more unstable than the performance of the sample processed under the phase transition temperature, resulting in a decrease in hardness. According to the measured H, the specific indentation depth is shown in Table 2. From Table 2, the processing temperature at 850 °C, the indentation depth got the minimum value. And the indentation depth at this time is smaller than the thickness of the hardened layer. From the Table 2, it can be seen that when the processing temperature is 750 °C, the indentation depth is the largest, and this indentation depth is greater than the thickness of the hardened layer, so the measured H is greatly affected by the Ti substrate, resulting in the surface at this time got to the low hardness. When the processing temperature is 950 °C, although the indentation depth is very large, it can be seen from the results of Figure 7 that the hardened layer peels off. This is the reason for the low the H at this processing temperature.

**Table 2.** Indentation depth with different processing temperatures.

| Processing Temperature/°C | Untreated | 750 | 800 | 850 | 900 | 950 |
|---|---|---|---|---|---|---|
| Indentation depth/μm | 5.47 | 3.15 | 2.57 | 2.43 | 2.65 | 2.99 |

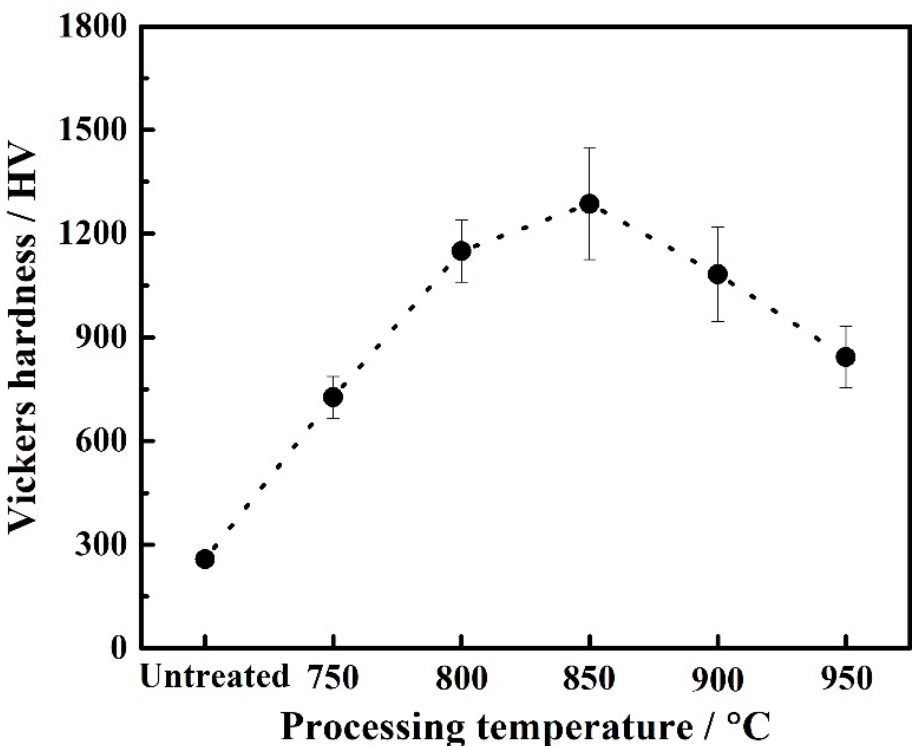

**Figure 5.** Surface hardness of samples with different processing temperatures.

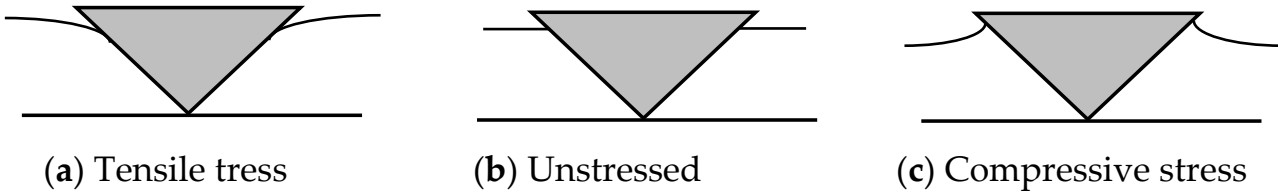

**(a)** Tensile tress    **(b)** Unstressed    **(c)** Compressive stress

**Figure 6.** Theoretical surface status around the contact for indentation states.

Describe later, another reason for the higher hardness at 850 °C is that a large number of whiskers are produced at this processing temperature. The whiskers have higher hardness, resulting in higher the H at this processing temperature.

During the surface hardness testing, the residual stress on the sample surface will interact with the force introduced by the indenter, which will affect the actual indentation morphology. Based on the surface hardness indentation morphology, the properties of the residual compressive stress on the sample surface can be qualitatively determined. Based on these results, it is helpful to explain the wear resistance of the surface layer. Appropriate residual compressive stress on the surface is conducive to the improvement of wear resistance. According to pieces of literature [23–25], when measuring the H, the residual stress will affect the shape and size of the sample indentation mark. When there is tensile stress on the surface of the sample, dimples will occur around the indentation mark, and when there is compressive stress, protrusions will occur around the indentation mark. The specific form can be seen in Figure 6. The SEM micrographs of the indentation morphology in this experiment with different processing temperatures were shown in Figure 7. It is assumed that the residual stress of the untreated sample is about zero. Based on the results of Figure 6, the type of stress in Figure 7 can be determined. Figure 7b is the tensile stress and the Figure 7c–f is the compressive stress. It can be seen from Figure 7c–f that as the processing temperature increases, the compressive stress is different. When the processing temperature increased, the compressive stress increased first and then decreased gradually. Especially when the temperature is 900 °C, due to the large compressive stress

between the hard layer and the substrate, cracks appear on the surface (Figure 7e). In addition, when the temperature increased to 950 °C, it can be seen from Figure 4 that the hardened layer peeled off, so no obvious cracks were found on the surface.

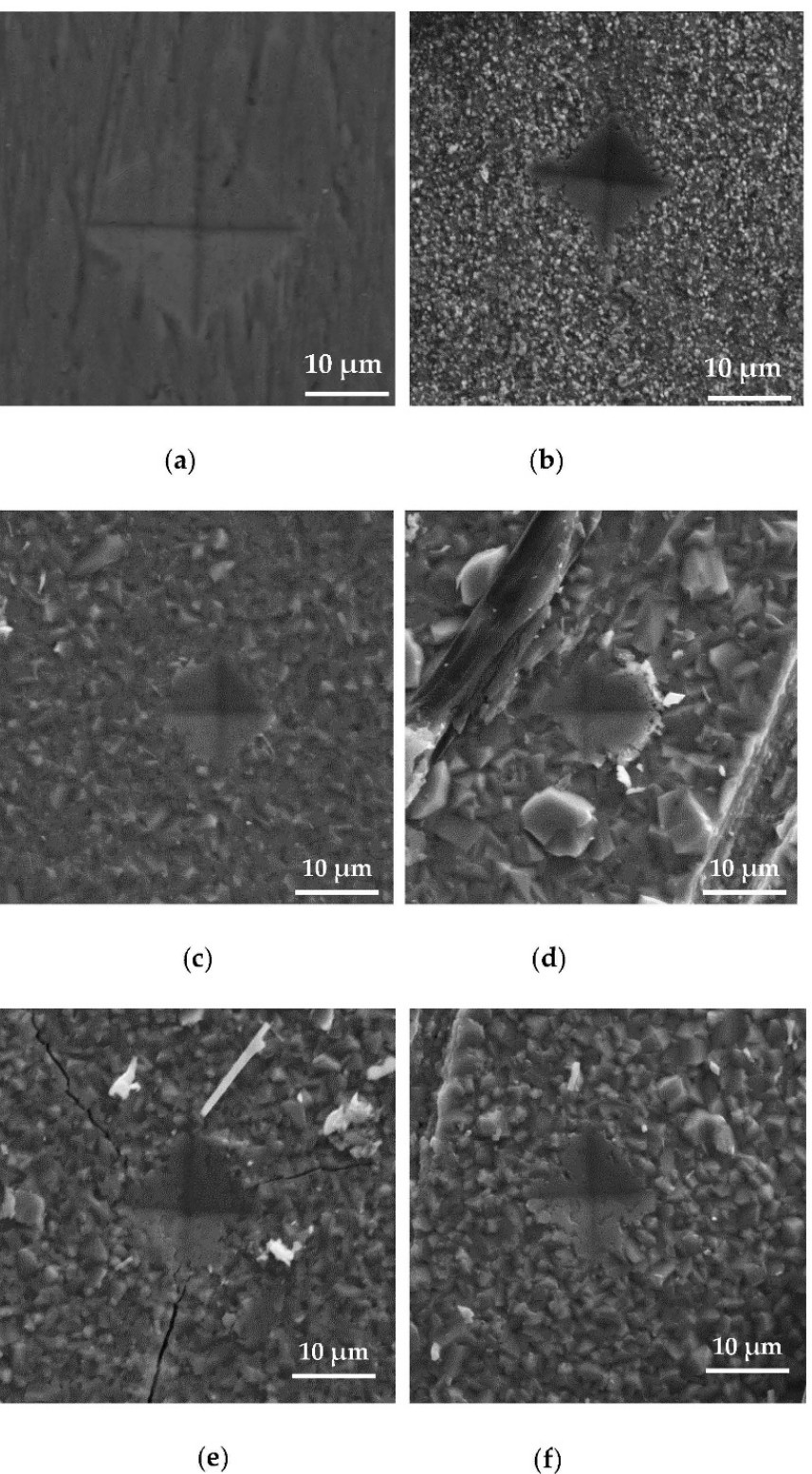

**Figure 7.** SEM micrographs of the impress with different processing temperatures: (**a**) untreated and (**b–f**) 750 °C, 800 °C, 850 °C, 900 °C, 950 °C.

In the indentation analysis, Figure 8 shows Young's modulus (E) and indentation depth curves of samples with different processing temperatures. The loading time, holding time, and unloading time of the indentation tester are all 5 s. The E characterized the ability of materials to resist elastic deformation. The indentation modulus of pure titanium was about 122 GPa. The indentation modulus of samples after processing was higher than the untreated sample, but when the processing temperature was higher, the value was lower than the untreated sample. From Figure 8, when the processing temperatures increased, E increased first and then decreased gradually. The modulus of elasticity can be regarded as an index to measure the difficulty of elastic deformation of the material. The larger the value, the greater the stress that causes the material to undergo a certain elastic deformation. The greater the material stiffness, the elasticity occurs under certain stress, the smaller the elastic deformation. Based on this, when the processing temperature is 850 °C, the E reaches its maximum value in this paper. This result is also mentioned in Ref. [26]. It means that the stiffness is the largest and the elastic deformation is the smallest in the design processing temperature [26]. In addition, the H also increased first and then decreased, as shown in Figure 5. When the processing temperature is 850 °C, E reaches the maximum value, but the error fluctuation range is relatively large at this processing temperature. This is due to a large amount of carbon fiber on the surface, which affects the measurement results.

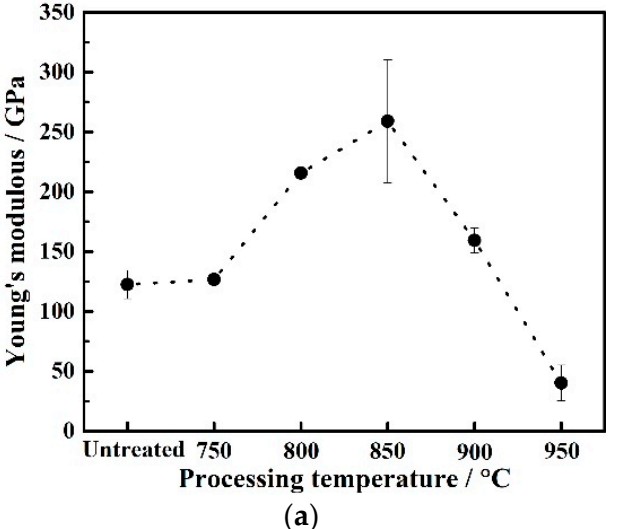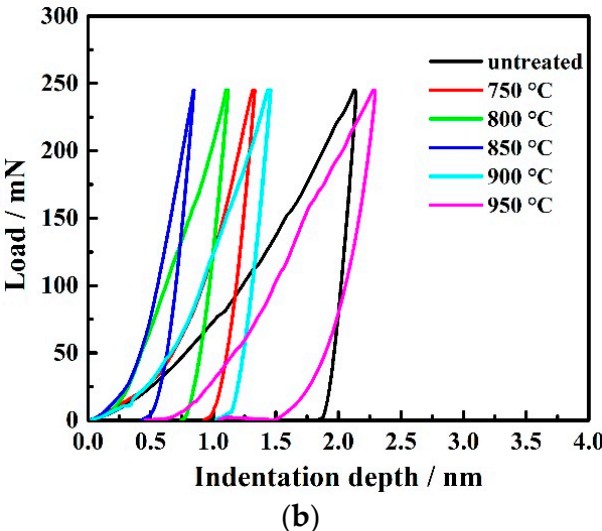

**Figure 8.** Young's modulus (**a**) and indentation depth curves (**b**) of samples with different processing temperatures.

The indentation results reveal that both the elastic modulus and the hardness increase with the decreasing residual stress of the samples. When the E was large, the amount of elastic recovery was large during the per unit time. It was also indicated that the sample has better resistance to plastic deformation at 850 °C. It has resulted that the indentation modulus reached the maximum value at this processing temperature. From Figure 8, when the processing temperature at 950 °C, the curve was not smooth and a polyline appeared. It is due to cracks occurred in the hardened layer and the hardened layer peeled off at this processing temperature. This result can be observed in Figure 4e.

### 3.3. Tribological Properties

The SEM micrographs and EDX analysis of surfaces with different processing temperatures are shown in Figure 9. After processing, the surface of the sample is cleaned with a nylon brush. The remaining carbon fiber content of the surface is shown in Table 3. As shown in this table, when the processing temperature increased, the carbon fiber content increased first and then decreased. In addition, when the processing temperature is 850 °C, the carbon fiber content got the maximum value. It shows that at this processing temperature,

the carbon fiber content remaining on the surface is the largest. As shown in Figure 9, when the processing temperature at 750 °C, it is almost no carbon fiber on the surface. During the processing temperature increased to 900 °C, the content of carbon fiber on the surface is less. In addition, there are traces of carbon fiber peeling off the surface. It is due to the decrease in the adhesive between the carbon fiber and the surface, causing the carbon fiber to peel off. When the processing temperature is 950 °C, although there are carbon fibers on the surface, almost all of them have peeled off. It can be seen from Figure 9 that in addition to carbon fibers, there are also white substances on the surface of some samples. As shown in Figure 9, when the processing temperature is 750 °C, the white substance does not exist on the surface. From the subsequent experimental results shown in Figure 10, it can be seen that the white substance is caused by the phenomenon of whisker growth. The driving force of the whisker growth phenomenon is the existence of compressive stress [27–29]. When the processing temperature is 750 °C, there is tensile stress. At this processing temperature, whisker growth does not occur, and there is no white substance. The specific discussion can be seen from the following discussion. As the processing temperature increases, the content of the white substance increased first and then decreased. Base on this phenomenon, perform magnification (red circle position) and point element analysis on Figure 9c, and the results are shown later in present research report.

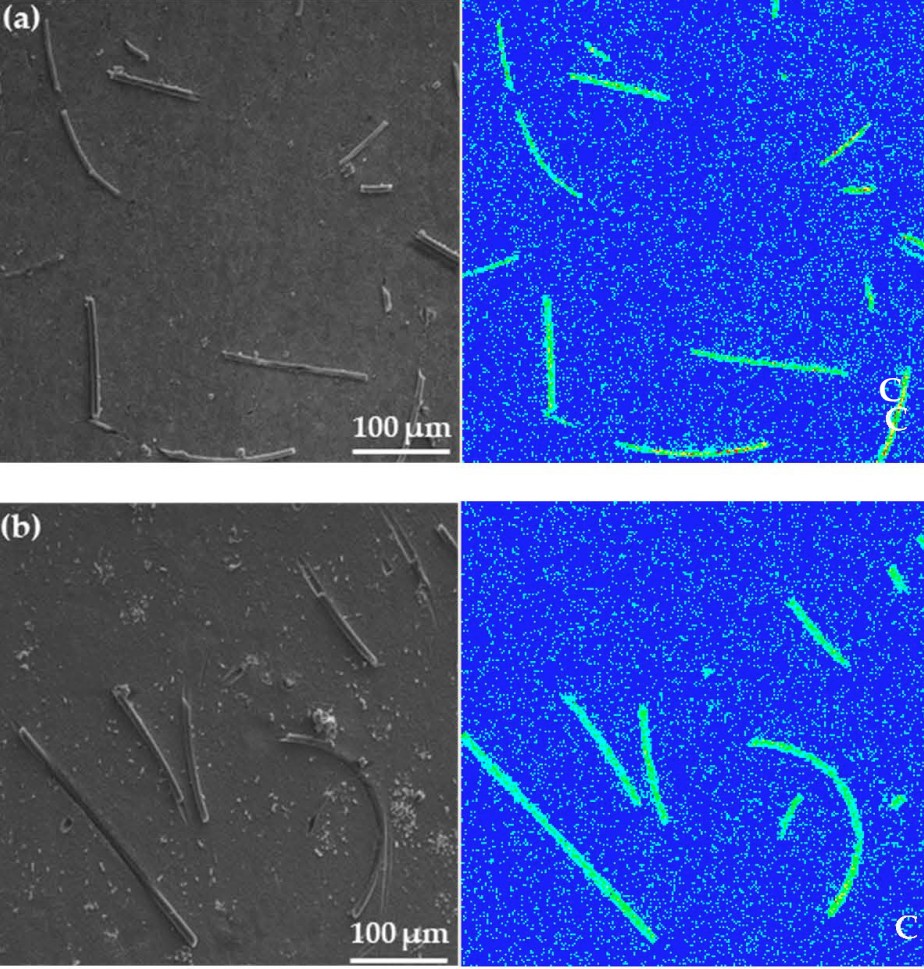

**Figure 9.** *Cont.*

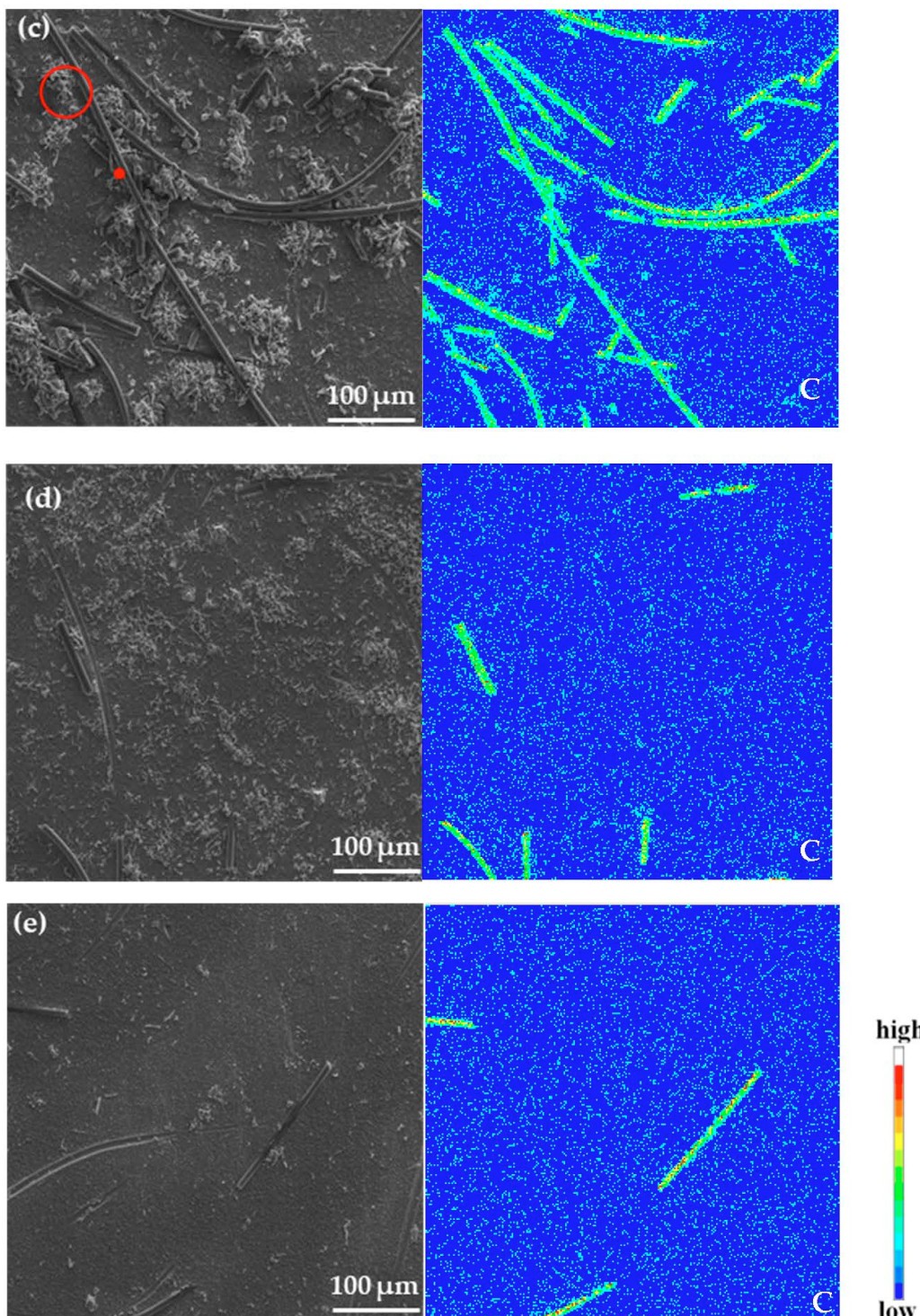

**Figure 9.** SEM micrographs and EDX analysis (C represents carbon element) of the treated sample's surfaces with different processing temperatures: (**a–e**) 750, 800, 850, 900, and 950 °C. (Amplify and analyze the elements marked in red circle and point. The results are shown later.).

**Table 3.** Remaining carbon fiber content of surface with different processing temperatures.

| Processing Temperature/°C | 750 | 800 | 850 | 900 | 950 |
|---|---|---|---|---|---|
| Carbon content/% | 2.81 | 3.37 | 8.35 | 1.77 | 1.32 |

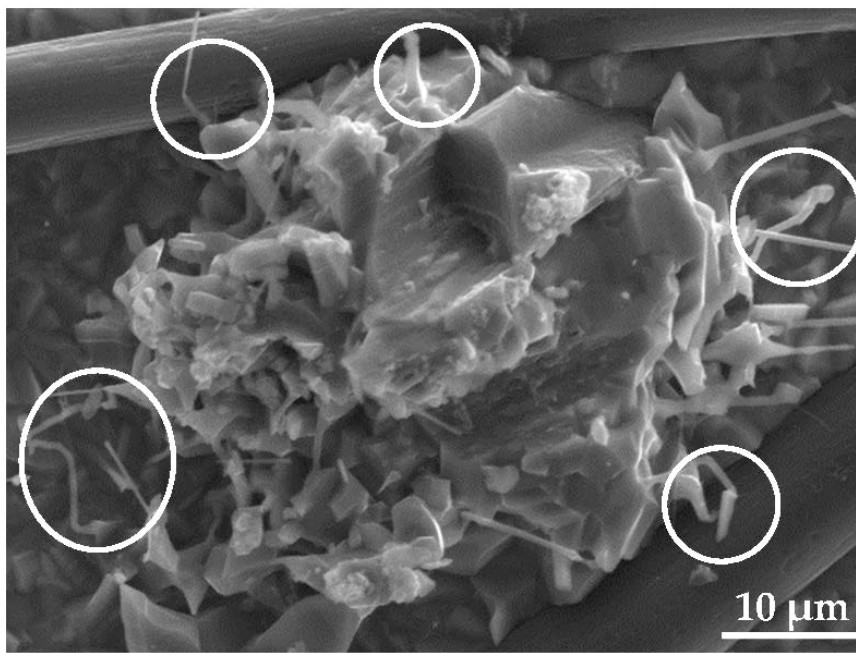

**Figure 10.** SEM micrographs of the surfaces with the processing temperature at 850 °C. (The white circle position is the growing whisker.).

A SEM micrograph of white substances on the surface with a processing temperature of 850 °C is shown in Figure 10. It can be seen from Figure 10 that there are many growing whiskers (the white circle position). This is a phenomenon of whisker growth that occurs at high temperatures [27]. The driving force for growth is thought to be the compressive stress of the hardened layer. This is also discussed in [28,29]. Due to the presence of compressive stress, substances are prone to whisker growth in the case of rapid growth. The rapid growth of matter is due to the formation of surface oxides and atomic diffusion. At low magnification, the surface is full of white substances. This is also the reason why the residual compressive stress of the sample is different as the temperature increases, resulting in the different white substance content on the surface during the processing temperature range from 800 to 950 °C. In addition, when the processing temperature is 750 °C, there is no white substance on the surface because the residual stress is tensile stress, so whisker growth does not occur. When the processing temperature is 850 °C, the whisker growth phenomenon is most obvious. This is because the compressive stress at the processing temperature reaches its maximum value. In order to relieve the stress's influence on the sample, the phenomenon of whisker growth has occurred. In other words, the phenomenon of whisker growth can play a role in alleviating stress.

In addition, the elemental analysis of the white substance is also performed, and the results are shown in Table 4. From Table 4, carbon, oxygen, and titanium were also detected. White substances are considered to be oxides. After processing, the surface of the sample is cleaned with a nylon brush. The nylon brush drives the carbon fibers on the surface to slide, which is why the white substance contains carbon. When oxygen diffuses into titanium to form surface oxides, under the action of stress, whisker growth occurs. It is precise because of the presence of high-hardness whiskers that prevent the carbon fiber from peeling off. It leads to the carbon fiber content remaining on the surface reaches the maximum with the processing temperature is 850 °C. In addition, Zhou et al. [30] found that whiskers have an stress-alleviating effect. There are whisker needles in all directions in space, which can better prevent the generation and development of cracks, which are not available in other powder-like materials. When the processing temperature increased to 900 °C, there is less whisker growth on the surface. Based on that reason, cracks are easily generated. This result can be seen in Figure 7e.

**Table 4.** Element content of surface with the processing temperature at 850 °C.

| Elements | C | O | Ti |
|----------|------|-------|-------|
| wt/% | 9.44 | 38.55 | 52.01 |

In order to observe the presence of carbon fibers on the surface, the cross-section of the sample was photographed with the SEM. The tilt angle of the sample stage was set to 13° while taking images, and the results of the carbon fiber cross-section with the processing temperature at 850 °C with different locations are shown in Figure 11. As shown in Figure 11, it can be found that the carbon fiber penetrates the surface (the red circle position). It shows that it has a good adhesive with the hardened layer, and it is not simply attached to the surface. Based on these results, it can be thought that the content of carbon fiber on the surface may affect the tribological properties of the treated samples. During the wear testing, the carbon fiber exists between the grinding ball and the test sample in the form of rolling, which plays a role of lubrication.

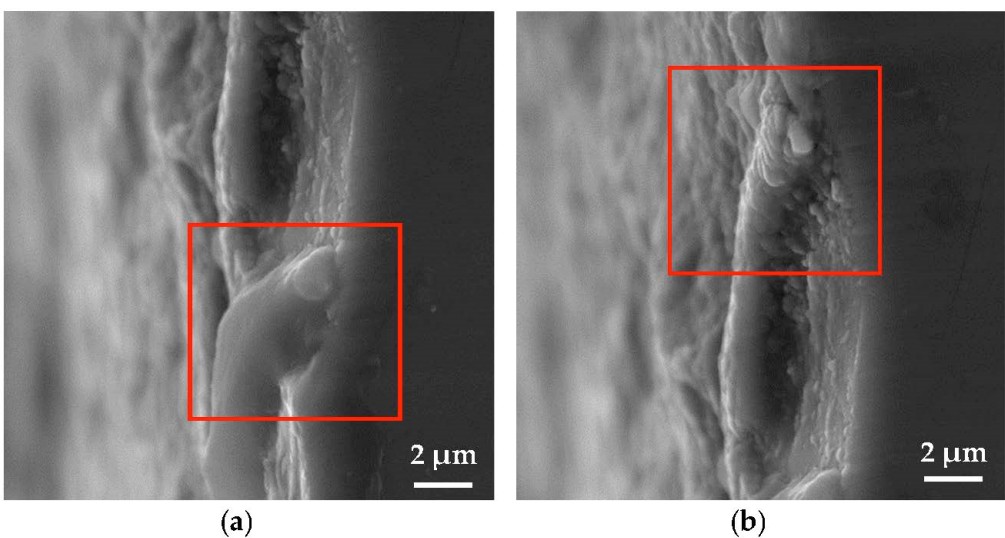

(a) (b)

**Figure 11.** SEM micrographs of the carbon fiber cross-section with the processing temperature at 850 °C with different locations (**a**,**b**). (The red box position is the carbon fiber that penetrates the surface position).

As shown in Table 5, the average wear depth of pure titanium was about 53.5 μm. The wear depth of treated samples was less than the untreated sample. When the processing temperature increased to 850 °C, the wear depth reached the minimum value. This is related to the hardness of the hardened layer. The higher the hardness due to the smaller the wear depth of hardened layer. The wear depth can be calculated by the next Equation (8) by supposing that an abrasion partner ball was worn [31]:

$$d = r - \left[ (r)^2 - \left(\frac{w}{2}\right)^2 \right]^{\frac{1}{2}} \tag{8}$$

In Equation (8), $r$ = 2380 μm, $d$ and $w$ represent a calculated value of the wear depth and a measured value of wear width.

With the increasing processing temperatures, the wear depth and the wear width decrease first and then increase gradually. When the processing temperature increases to 850 °C, the wear width and wear depth reach the minimum values of 385 μm and 2.01 μm, respectively. As shown in Table 5, the calculated value of the wear depth exceeded the measured value. It was found from these results that the abrasion of the ball surface was

caused in addition to the surface of the diffusion layer. It was inferred that the cause of these disagreements is changed from mechanical wear to abrasive wear.

**Table 5.** Average wear width and wear depth of samples with different processing temperatures after wear testing.

| Processing Temperature/°C | Wear Width/μm | Wear Depth/μm | |
|:---:|:---:|:---:|:---:|
| | | **Measured** | **Calculated** |
| untreated | 1240 | 53.5 | 82.2 |
| 750 | 1179 | 11.8 | 74.2 |
| 800 | 537 | 3.97 | 17.3 |
| 850 | 385 | 2.01 | 7.8 |
| 900 | 793 | 16.0 | 33.3 |
| 950 | 1024 | 19.4 | 55.7 |

The 2D profilometric views of both untreated and treated samples are shown in Figure 12. From the 2D profiles showing the wear depths, when the processing temperature is 850 °C, it is clearly seen that the wear resistance of the untreated sample is improved substantially. A wide and deep wear track develops on the untreated sample after wear testing. The rough 2D wear track was seen in Figure 12a. The untreated sample shows a larger and deeper wear track compared to treated samples. Severe plastic deformation with heavily smearing and scratches occur on the surface of untreated samples during wear testing. Comparing the measured wear depth value of treated samples and the untreated sample, when the processing temperature at 750 °C, the wear track was deeper, but it was lower than the untreated sample. It is due to the increasing hardness reduced the impact on the sample when the ball wore it down. When the processing temperature is 750 °C, the hardened layer is very thin, and micro-particles will be generated that increase the wear. This consideration is supported by the fact that many small particles on the edge of the wear scar were observed in the wear scars observation results in Figure 12. When the processing temperature is 850 °C, the worn surfaces shows very limited deformation and smearing, and the wear depth is decreased significantly, which is related to the hardened layer formed. The hardened layer plays a role in protecting the substrate Ti. As can be seen from Figure 4, the hardened layer at this processing temperature is the thickest and it can be clearly seen that the wear depth was shallower when the hardened layer was thicker. This extraordinary improvement in wear resistance of the treated sample is attributed to the hard surface layer formed during processing (Figure 4). In addition, it can be seen from the 2D profilometric view at this processing temperature that the surface is uneven. This is due to a large amount of carbon fiber existing on the surface. As pointed out above, a surface layer with ultra-high hardness was achieved on the top-most layer with the substrate having gradually decreasing hardness. Such surface formation is beneficial, especially for material wear resistance because the hard surface layer increases both the adhesive and abrasive wear resistance of the soft Ti substrate. The layer beneath just below the top-most layer with relatively low hardness may have a good damping effect which prevents delamination during rubbing, especially under heavy loading conditions. When the processing temperature is higher than 850 °C, the wear becomes serious. Specifically, when the processing temperature at 900 °C, the wear width and wear depth reached 793 μm and 16.0 μm, respectively. In addition, the wear width was wider and there is obvious adhesion on the surface of the wear scar. Moreover, it can be seen from the results in Table 5, that when the processing temperature is increased to 950 °C, the width of the wear scar becomes wider and the depth of the wear scar becomes deeper. The maximum values of 1024 μm and 19.4 μm are reached, respectively. This indicates that the wear has become severe on the surface of the hardened layer. It is a result that the hardened layer peeled off and generates many particles on the surface lead to abrasive wear become serious.

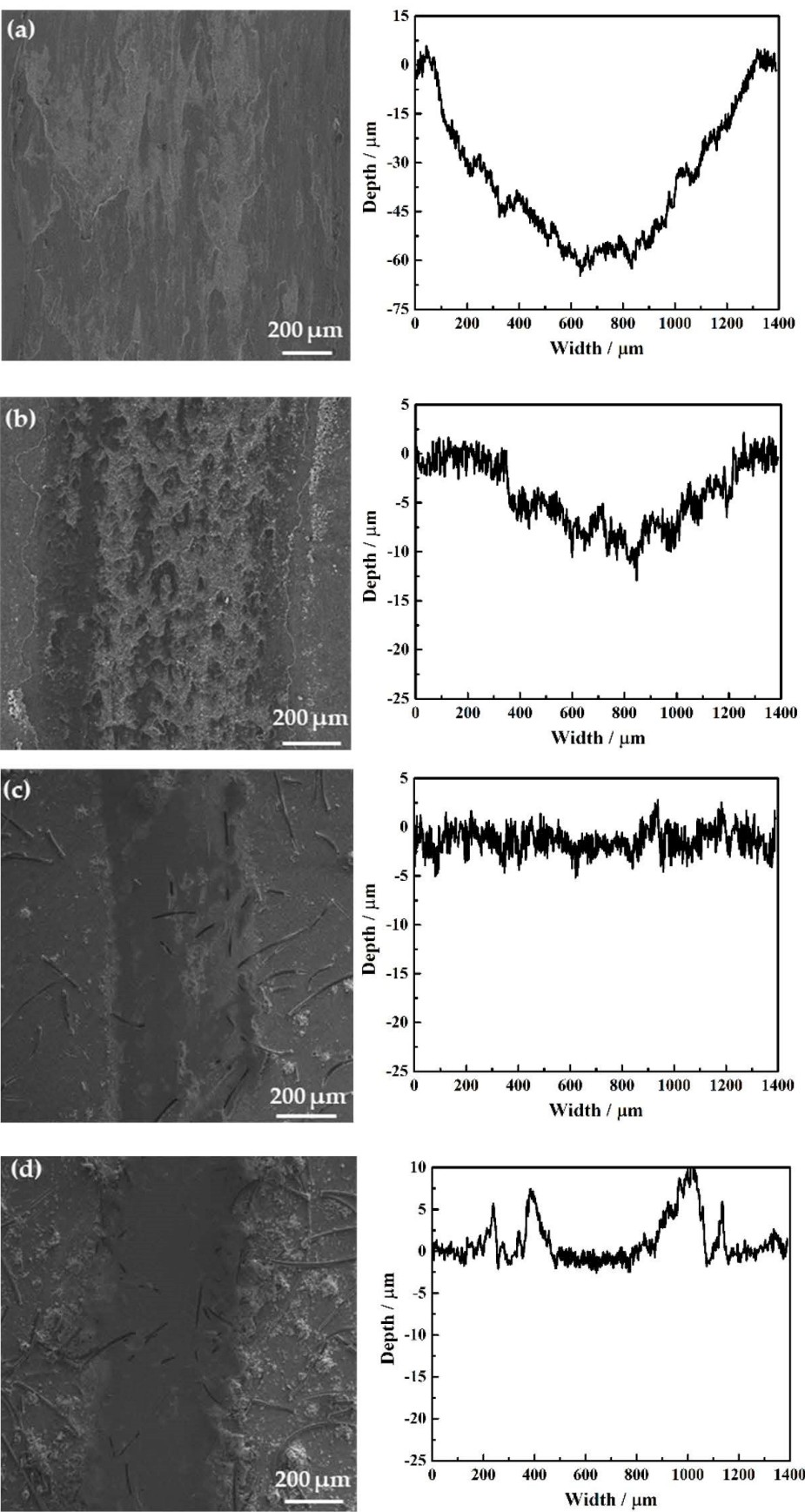

**Figure 12.** *Cont.*

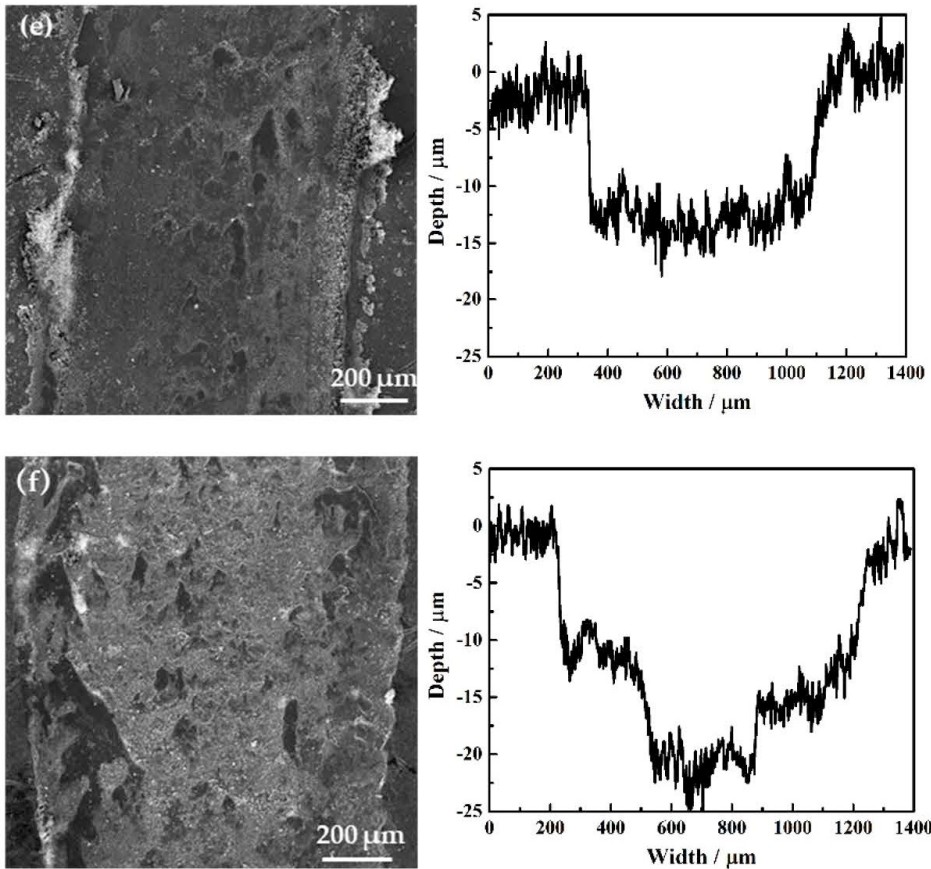

**Figure 12.** SEM micrographs, EDX analysis and 2D profilometric views of the surfaces of the (**a**) untreated sample, and those prepared with different processing temperatures after wear testing: (**b**–**f**) 750, 800, 850, 900, and 950 °C.

It can be seen from these results that the main form of wear during the experiment is adhesive wear. At high processing temperatures, a large number of oxides (white substances) are produced on the surface. The existence of oxides plays a role in protecting the matrix. It can improve the wear resistance of the samples. In the wear testing, the oxide peeled off from the surface of the sample and adhered to the grinding ball. As the wear time increases, there are more adherents on the grinding ball, resulting in serious wear. Especially in Figure 12a,b,e,f), this phenomenon can be seen. However, as shown in Figure 12c,d, the adhesion phenomenon was not clearly observed. This is because many carbon fibers are attached to the surface of the treated samples, the carbon fibers are peeled off during the wearing testing, leaving dimples on the surface. The peeled carbon fiber exists between the grinding ball and the surface of the sample in the form of rolling, which plays a role in rolling lubrication. As wear testing time increases, part of the wear debris enters the dimple, which plays a role in inhibiting abrasive wear. The specific results are shown later in this paper.

In addition, Zhou et al. [30] also found that whiskers can play a role in reducing wear. Whiskers have a fibrous structure. In the wear testing, they play a lubricating role and reduce wear. Zeng et al. also showed that $Mg_2B_2O_5$ nanowires that produce the phenomenon of whisker growth can be used as excellent anti-wear additives as the load increases and the wear resistance improves [32]. This is also one of the reasons why the wear width and the wear depth reach the minimum values 385 μm and 2.01 μm respectively, which makes the wear resistance at the processing temperature is 850 °C reach the optimal during the designed processing temperature.

The enlarged SEM micrographs of wear marks at 800 °C and 850 °C are shown in Figure 13, where it is found that the carbon fibers inside the wear marks were peeled off from the surface. Compared with the processing temperature of 800 °C, the carbon fiber content on the surface at 850 °C is more, resulting in more dimples left during the wear testing. As shown in Figure 13, debris exists in the dimples (the red circle is where the worn debris exists). Compared with 800 °C, when the processing temperature is 850 °C, the probability of wear debris appearing in the dimples is larger, indicating that the effect on inhibiting abrasive wear is greater at this processing temperature. This can play a wear resistance role during the wear testing. This is considered to be one of the causes of the low wear depth and wear width seen at 850 °C.

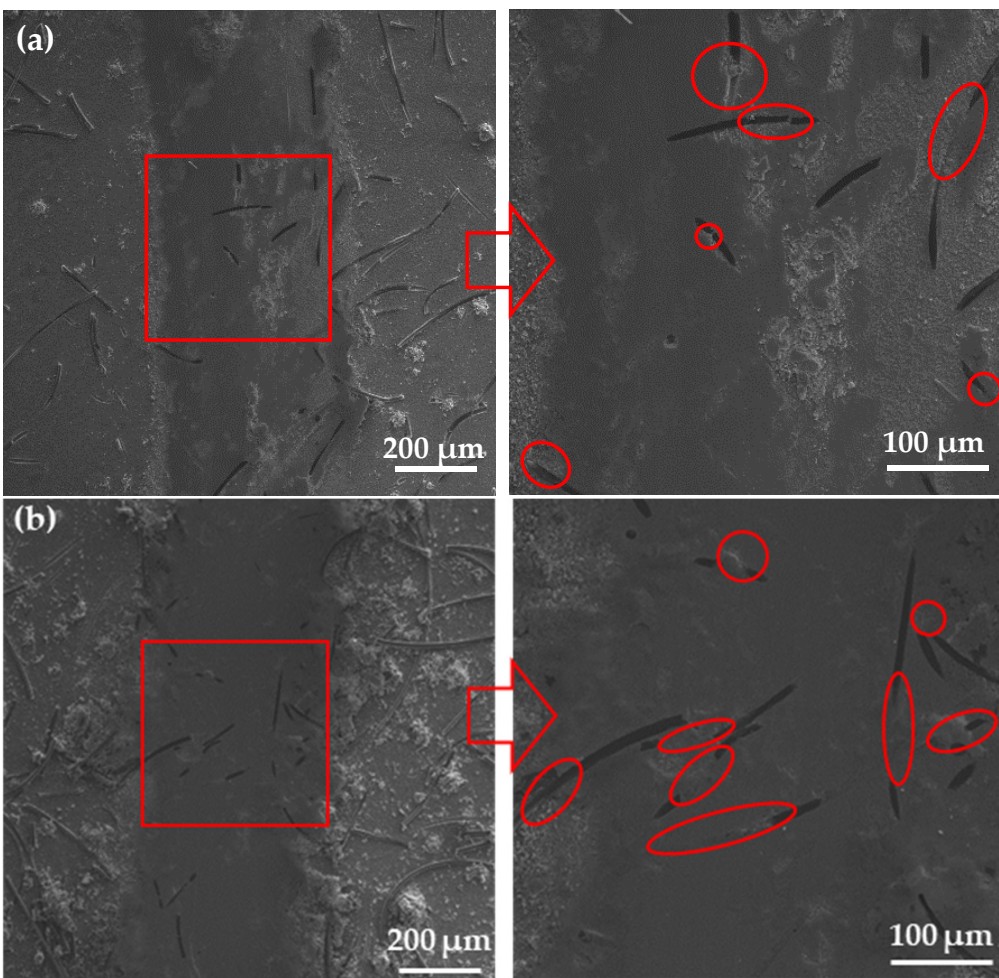

**Figure 13.** SEM micrographs of carbon fiber in wear scar with processing temperatures at (**a**) 800 °C and (**b**) 850 °C. (Enlarge the position of the red box, and the result is shown in the figure on the right. The red circle is where the worn debris exists.).

Figure 14 shows the trend of the coefficient of friction (COF) and fricative average value with different processing temperatures during the wear testing. As shown in Figure 14, the COF and fricative average value of the untreated sample higher than treated samples with different processing temperatures. This shows that the sample treatment has the effect of reducing the fricative average value and the COF. Besides, with the increasing processing temperatures, the COF and fricative average value decreased first and then increased gradually. Among the different processing temperatures studied in this article, the fricative and COF average value reached the minimum value at 850 °C. This is the result that the fact the thickness of the hardened layer is the thickest, and at this temperature the H reached the maximum value, and the hardened layers were hard to peel off

during the wear testing. When the processing temperature increased to 950 °C, the COF and fricative average value had a high value. This is due to the fact the thickness of the hardened layer is thinner, the hardened layer peels off and changes into particles. At this time, the particles wear with the sample and the grinding ball, which generates a large fricative effect. Furthermore, particles exist on the surface of the sample and the grinding ball. During the wear testing, this increases the COF.

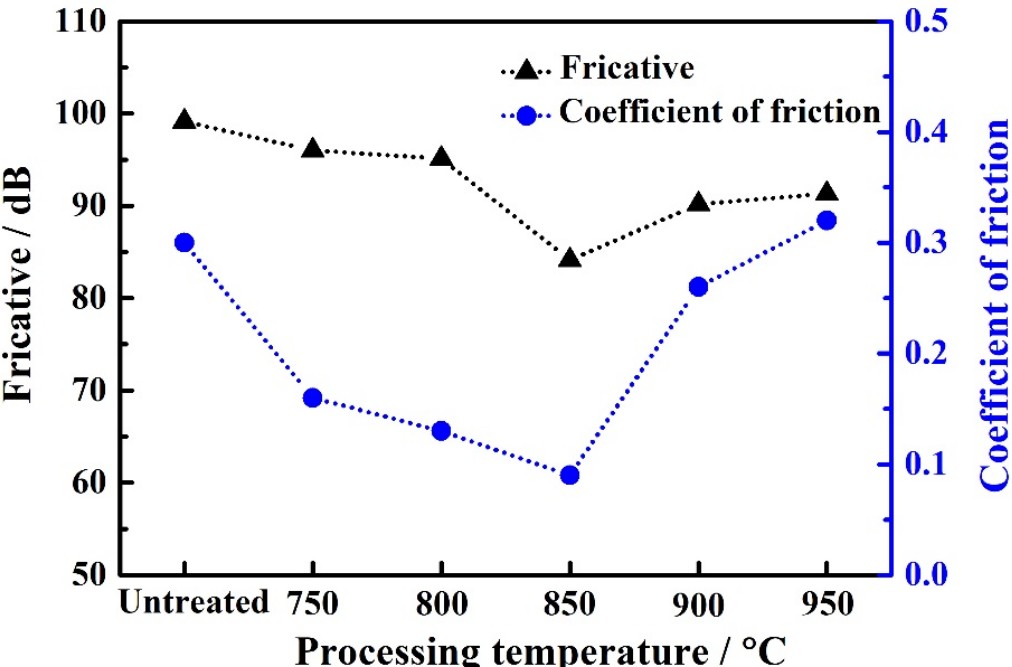

**Figure 14.** Trend of coefficient of frication and fricative average value with different processing temperatures during the wear testing.

In addition, whiskers have a fibrous structure, and are more likely to deform when subjected to external forces, and can absorb shock and vibration energy, thereby reducing noise. Therefore, when the processing temperature is 850 °C, the whisker growth phenomenon is most obvious, resulting in the minimum noise observed at this processing temperature.

Figure 15 shows the variation of the COF with the wear testing time under different processing temperatures. It can be seen from the figure that the COF of the untreated sample is higher than that of the treated sample, indicating that the processing method in this article reduces the COF As shown in Figure 15, when the processing temperature is 850 °C, the curve is the flattest. This is due to the presence of a large number of oxides, carbon fibers, and whiskers on the surface. At the beginning of the wear testing, due to the rolling lubrication of the carbon fiber and the presence of a large number of whiskers on the surface, the friction coefficient was reduced. As the processing time increases, the carbon fiber will peel off. Because the whiskers have a fibrous structure, the wear is slowed down and the friction coefficient is reduced. When the experiment time is gradually increased, the carbon fibers and whiskers on the surface are completely destroyed. At the same time, the grinding ball comes into contact with the hardened layer on the surface and wears. The presence of oxides plays an important role in protecting the matrix and reducing wear.

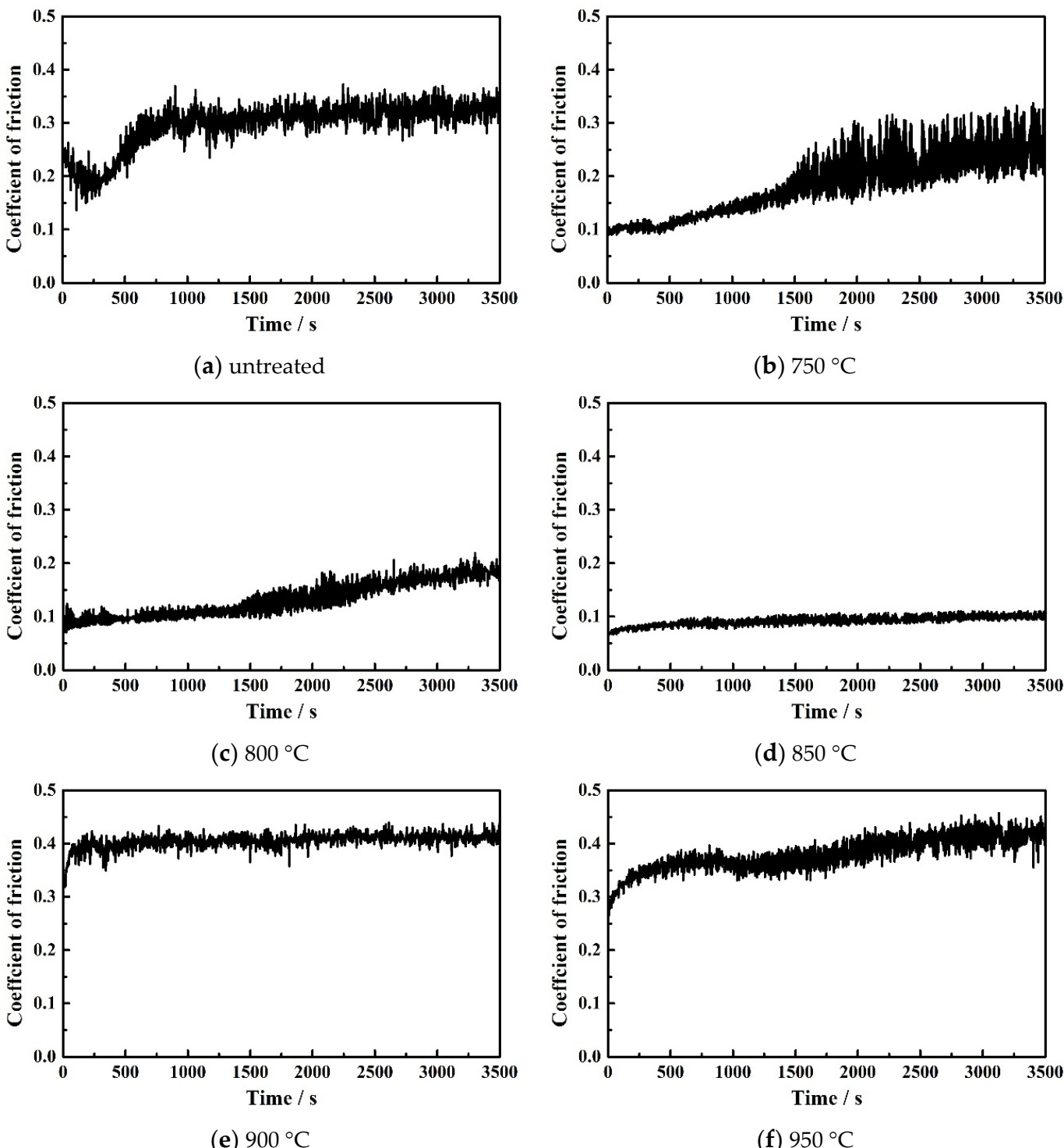

**Figure 15.** Coefficient of friction variation with the wear testing time at different processing temperatures for the (**a**) untreated sample, and (**b–f**) 750 °C–950 °C.

## 4. Conclusions

The effect of the oxidation processing temperature on the formation of the hardened layers on grade-2 pure Ti by using carbon sheet was investigated. Detailed characterization of the structure, mechanical and tribological properties yielded the following conclusions:

(1) Oxidation at all tested temperatures resulted in the formation of a hardened layer, and as the processing temperature increases, the thickness of the hardened layer

increases and then decreases. The relative thickness depends greatly on the processing temperature, with reaches a maximum thickness at a temperature of 850 °C.

(2) Severe plastic deformation with heavy smearing and scratches occurs on the surface of untreated samples during wear testing. The worn surfaces of the oxide samples showed minor deformation and smearing effects at high processing temperatures, with minimum wear depth and width for a processing temperature of 850 °C.

(3) This processing reduced the fricative value and coefficient of friction. As the processing temperature increases, the fricative value of the samples decreases initially and then increases. When the processing temperature is 850 °C, the fricative value and coefficient of friction reached their minimum values, respectively.

(4) The process used in the present study substantially increased the wear resistance of Ti due to the high resulting surface hardness. The reason why the wear resistance is improved by processing is not only the presence of oxides but also the presence of carbon fibers. Besides, the phenomenon of whisker growth also plays a role in reducing wear.

**Author Contributions:** Conceptualization, T.C. and S.K.; methodology, S.K.; software, S.N.; validation, T.C., S.K. and S.N.; formal analysis, T.C.; investigation, T.C.; resources, S.N. and L.Y.; data curation, T.C.; writing—original draft preparation, T.C.; writing—review and editing, S.K.; visualization, S.K.; supervision, L.Y. All authors have read and agreed to the published version of the manuscript.

**Funding:** This research received no external funding.

**Institutional Review Board Statement:** Not applicable.

**Informed Consent Statement:** Not applicable.

**Acknowledgments:** We are pleased to acknowledge experimental support by Shinji Koyama, Shinichi Nishida and Lihua Yu, thanks the Functional Interface and Surface Fabrication laboratory at Gunma University, where the samples where produced and macroscopically compressed as part of a different research project.

**Conflicts of Interest:** No potential conflict of interest was reported by the authors.

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
