# Peer review of "Influence of Oxidation Processing Temperature on the Structure, Mechanical and Tribological Properties of Titanium Using Carbon Sheets"

_metals, doi:10.3390/met11040585_

Round 1
Reviewer 1 Report
The literature review in the abstract should be expanded, discuss what previous authors did and what were their main findings and how does your current study bring new knowledge and benefit this field. What is the research gap did you find from the previous researchers in your field? Mention it properly. It will improve the strength of the article.
Rename experimental section to materials and method section, also add some images and figures for the equipment and tests setup used in this study since this is an experimental study it is important to clearly show all details of the experimental work
“It was inferred that oxygen diffused at high temperatures and became more pronounced as processing temperature increasing.” Why? Explain this phenomena and support with references
“This can indicate that the carbon atoms in the carbon fiber have reacted with oxygen in the air, causing the mass to decrease” this claim needs to be supported with a reference, what about past studies what did they find, is it same or different from your findings, discuss more in details and support with references
The authors provide good explanation to what they observe but they need to critically explain it and support it with references and compare it with previous studies to highlight similarities and differences then explain them
“Furthermore, oxidation is a process in which oxygen atoms diffuse from the outside to the inside and react with titanium. With increasing the processing time, the content of oxygen on the surface and inside of the sample is different” this needs referencing again.
“Besides, the thickness of this dark contrast layered structure tended to increase with increasing processing temperature.” So what does this mean for the overall process and results? Please explain and support with references
“From these results, it was due to high-temperature embrittlement” how do know that for sure? Please explain more and support with references
“This suggests that the diffusion of C in β-Ti was faster than the diffusion of C in α-Ti. B” this needs to be further discussed and supported with references. Could it be something else as affection the results?
“However, the hardness of the hardened surface is lower than the hardness” so what does that mean, is it good or bad for the overall material properties? Please don’t just state some findings without explaining them in more details and supporting with references
“Young's modulus increased first and then decreased gradually.” What does that mean? Is it good or bad, also why this happened, please explain in more details all the phenomena and observations you find and support with references
“It may be due to cracks occurred in the hardened layer and the hardened layer peeled off at this processing temperature” ok here you make a new speculation but you don’t mention what past studies found, is it similar to yours or different, you need to discuss in more depth your results and findings.
“When the processing temperature is 750°C, the white substance does not exist on the surface.” This claim needs a reference and more explanation, why this happened?
“which may be related to the hardened layer formed.” Again so many speculations, please explain further and support with references, also is this the only possible reason or there might be other ones responsible for the hardened layer?
“wear became serious, and obvious” this is a very vague sentence! Define very serious here, it is better to quantify any findings like wear to give clear indication of what is meant by serious here compared to something not serious
“it played a lubricating effect and reduced wear.” Again another speculation but not further explanation or comparing with past studies..
“it played a lubricating effect and reduced wear.” By how much? It is not clear to me the extent of its effect on the lubrication phenomena.
The results are merely described and is limited to comparing the experimental observation. The authors are encouraged to include a detailed discussion which critically discuss the observations from this investigation with existing literature.
Author Response
Response to Reviewer 1 Comments
Point 1: The literature review in the abstract should be expanded, discuss what previous authors did and what were their main findings and how does your current study bring new knowledge and benefit this field. What is the research gap did you find from the previous researchers in your field? Mention it properly. It will improve the strength of the article.
Response 1: Thank the reviewer for the comments. Your suggestions were adopted and the relevant part of our manuscript was revised as follows:
Bansal R. et al. reported the study investigates the influence of oxidation temper-ature for pure titanium to achieve improve corrosion resistance [10]. Maytorena Sánchez A. et al. discussed the formation TiO2 coating can improve the hardness by oxidation [11]. Du H.L. et al. reported the high-temperature corrosion of Ti and Ti-6Al-4V [12]. It found that adding Al into the simple boronized coating is beneficial for the high-temperature oxidation resistance. Most of today's oxidation articles are about the oxidation of alloys. Articles about the oxidation of pure titanium are only a single study of its corrosion or mechanical properties, and there are few studies on the overall structure, mechanical, and tribological properties of pure titanium. In addition, based on Ref. 12, this paper proposes a conjecture about whether the use of carbon cloth will improve the performance of pure titanium during oxidation treatment, and conducts specific studies on the structure, mechanics, and friction and wear properties. Many researchers have research pure titanium preliminarily, but there are few reports about oxidation by using carbon sheet.
Point 2: Rename experimental section to materials and method section, also add some images and figures for the equipment and tests setup used in this study since this is an experimental study it is important to clearly show all details of the experimental work
Response 2: Thank the reviewer for the comments. Your suggestions were adopted and the relevant part of our manuscript were revised as follows:
The samples used in this study comprised a 50 × 20 × 5 mm3 plate cut from the as-received plate using fine cut machining (HEIWA TECHNICA, HS-45A2). Pure grade-2 Ti (composition in wt.%: 0.2 O, 0.003 N, 0.013 H, 0.25 Fe, 0.008 C, and remain-der Ti) was used as the substrate material. The surface to be treated of the Ti was fin-ished by grinding with emery paper (grade 4000). The pure titanium is heated by the air furnace in the atmosphere by using the carbon sheet. The schematic diagram of the sample setup was shown in Figure 1. The scanning electron microscope (SEM, HITACHI, S-3000N) image of the carbon sheet was shown in Figure 2. The titanium plate was placed between two pieces of carbon sheet and the load at 1 MPa was applied to it. Processing temperatures were changed from 750°C to 950°C for 2 h in the atmosphere. To determine the compound formed on the surface after processing, the treated sur-faces were investigated using an X-ray diffractometer (XRD, Rigaku, 2200VF) with CuKα radiation working at the optimum voltage of 32 kV and anodic current of 20 mA. Furthermore, the cross-section of treated samples was prepared by ion milling (HITACHI, IM4000) with 7200 s. Microstructural and morphological characteristics of hardened layers were examined using a SEM. Surface hardness testing was carried out throughout the hardened layers of the samples using a Vicker hardness tester (SHIMADZU, HMV-1) with the applied load is 0.98 N to gain the hardness of the hardened layer more accurately and at a dwell time of 15 s. In order to reduce the measurement error, the sample was measured for 5 times and averaged. Ball-on-disk dry sliding tests were performed at room temperature to evaluate the tribological properties of the samples using a tribometer (Rhesca, FPR-2000) with a zirconium dioxide (ZrO2) ball and a counter-face with a radius of 2380 μm. Samples with the dimensions of 50 × 20 × 5 mm3 were used for wear testing with a sliding linear speed of 200 mm/s, an applied force (loading force during the wear testing) of 4.9 N, and a test time of 3600 s (corresponding to a sliding distance of 720 m). The experimental apparatus of the wear and fricative testing was shown in Figure 3. All tests were performed at room temperature. The microstructural and morphological features of the wear track after wear testing were examined using SEM under secondary electron imaging mode at a voltage of 15 kV. The fricative value is measured by a condenser microphone (TENMARS, TM-103) installed 20 mm above the contact point between the ball and sample. The depth pro-file of the sliding portion after the wear testing was measured by a laser microscope. In order to reduce the measurement error, the sample was measured for 4 times and averaged.
Figure 1. Schematic diagram of the sample setup.
Figure 2. SEM image of carbon sheet.
Figure 3. Experimental apparatus of the wear and fricative testing.
Point 3: “It was inferred that oxygen diffused at high temperatures and became more pronounced as processing temperature increasing.” Why? Explain this phenomena and support with references
Response 3: Thank the reviewer for the comments. Your suggestions were adopted and the relevant part of our manuscript were revised as follows:
As shown in Figure 4, it was revealed that TiO2, Ti2C, and Ti8C5 were formed on the surface. It was inferred that oxygen diffused at high temperatures and became more pronounced with different processing temperature. Rosa, C. J. et al. discussed the oxygen diffusion in alpha and beta titanium in the temperature range of 932°C to 1142°C [13]. It found that the diffusion rate of oxygen in beta titanium is faster than that in the alpha titanium at the same processing time. For beta titanium, the diffusion rate becomes slower as the processing temperature increases.
Point 4: “This can indicate that the carbon atoms in the carbon fiber have reacted with oxygen in the air, causing the mass to decrease” this claim needs to be supported with a reference, what about past studies what did they find, is it same or different from your findings, discuss more in details and support with references
Response 4: Thank the reviewer for the comments. Your suggestions were adopted and the relevant part of our manuscript were revised as follows:
In this paper, the weight loss analysis of the carbon fiber treated at the processing temperature of 850°C for two hours was researched. The results are shown in Table 1. In the absence of external wear, only the carbon sheet was subjected to heat treatment experiments in the atmosphere. After the processing, the weight of the carbon sheet has changed, which can be considered as an oxidation reaction of the carbon sheet with a certain substance in the atmosphere. Based on this results, it found that under the same processing time, the weight of carbon fiber is reduced. This can indicate that the carbon atoms in the carbon fiber have reacted with oxygen in the air, causing the mass to decrease. As shown in the reactions of (1) and (3). Refs. 16 and 17 also show that re-actions (1) and (3) can occur, generating carbon dioxide and carbon monoxide, respectively [16,17]. This confirms the hypothesis that carbon atoms react with oxygen atoms in the air.
Point 5: “Furthermore, oxidation is a process in which oxygen atoms diffuse from the outside to the inside and react with titanium. With increasing the processing time, the content of oxygen on the surface and inside of the sample is different” this needs referencing again.
Response 5: Thank the reviewer for the comments. Your suggestions were adopted and the relevant part of our manuscript were revised as follows:
Furthermore, oxidation is a process in which oxygen atoms diffuse from the out-side to the inside and react with titanium. With increasing the processing time, the content of oxygen on the surface and inside of the sample is different. This result is also mentioned in Ref. 18. Abuluwefa H. et al. also studied the diffusion of oxygen with different temperatures. It shows that as the processing time increases, the oxygen content inside the sample increases and reaches a compound that maintains a stoichiometric ratio [18].
Point 6: “Besides, the thickness of this dark contrast layered structure tended to increase with increasing processing temperature.” So what does this mean for the overall process and results? Please explain and support with references
Response 6: Thank the reviewer for the comments. Your suggestions were adopted and the relevant part of our manuscript were revised as follows:
As shown in Figure 7, the hardened layer varied with different processing temperature were observed in the vicinity of the surface. In each processing temperature, there was a layered structure with dark contrast different from the base metal near the sample surface. Besides, the thickness of this dark contrast layered structure tended to increase with increasing processing temperature. This shows that at each processing temperature, a new substance is formed on the surface of the sample. These new sub-stances appear dark contrast layered under the SEM. King M.K. et al. also showed that during the oxidation processing, a layer of dark layered substance was formed on the surface of the sample under the SEM [19]. With the increasing processing temperatures, the thickness of the dark contrast layer increased first and then decreased gradually. From the EDX analysis results, it was found that the C and O detection areas coincided with the layered structure that had a dark contrast different from the substrate near the sample surface. Therefore, it is considered that oxides were generated in the dark contrast region by oxygen diffusion and infiltration treatment.
Point 7: “From these results, it was due to high-temperature embrittlement” how do know that for sure? Please explain more and support with references
Response 7: Thank the reviewer for the comments. Your suggestions were adopted and the relevant part of our manuscript were revised as follows:
From Figure 7 (e), it was found that the hardened layer peels off and the content of C and O decreased significantly at this temperature. From these results, it was due to high-temperature embrittlement, the compounds formed on the titanium surface were prone to peel at 950°C. Chan, K. et al. researched the dynamic embrittlement and oxidation-induced cracking in superalloys. It found that oxidation-induced crack growth plays a leading role in metal embrittlement [20]. In addition, the existence of residual stress also leads to the hardened layer peel off, show the following discussion for details.
Point 8: “This suggests that the diffusion of C in β-Ti was faster than the diffusion of C in α-Ti. B” this needs to be further discussed and supported with references. Could it be something else as affection the results?
Response 8: Thank the reviewer for the comments. Your suggestions were adopted and the relevant part of our manuscript were revised.
From Figure 7 (e), it was found that the hardened layer peels off and the content of C and O decreased significantly at this temperature. From these results, it was due to high-temperature embrittlement, the compounds formed on the titanium surface were prone to peel at 950°C. Chan, K. et al. researched the dynamic embrittlement and oxidation-induced cracking in superalloys. It found that oxidation-induced crack growth plays a leading role in metal embrittlement [20]. In addition, the existence of residual stress also leads to the hardened layer peel off, show the following discussion for details. Transformation of α-Ti ⇆ β-Ti is known to occur around 880°C, α-Ti has a dense hexagonal structure, and β-Ti has a body-centered cubic structure is there. At processing temperatures of 750°C and 800°C, O and C were strongly detected on the surface of the test piece and tended to decrease gradually. On the other hand, when the processing temperatures is 800°C and 850°C, O and C were strongly detected from the sample surface to the entire layered structure. This suggests that the diffusion of O and C in β-Ti was faster than the diffusion of O and C in α-Ti. This result is also mentioned in Ref. [13] and [21]. Besides, it is considered that cracking occurred when cooling from 850°C or higher due to lattice transformation.
Point 9: “However, the hardness of the hardened surface is lower than the hardness” so what does that mean, is it good or bad for the overall material properties? Please don’t just state some findings without explaining them in more details and supporting with references
Response 9: Thank the reviewer for the comments. Your suggestions were adopted and the relevant part of our manuscript were revised as follows:
As shown in the XRD results in Figure 4, titanium oxide and titanium carbide were detected strongly on the surface of the samples as the processing temperature in-creased. Sivakumar Bose et al. revealed that the hardness of TiO2 is about 800 HV, and the hardness of TiC is about 3200 HV. This may be due to the presence of TiO2 and TiC causing the hardness to increase [22]. However, the hardness value measured in present research is less than the hardness value in the Ref. 20.
Point 10: “Young's modulus increased first and then decreased gradually.” What does that mean? Is it good or bad, also why this happened, please explain in more details all the phenomena and observations you find and support with references
Response 10: Thank the reviewer for the comments. Your suggestions were adopted and the relevant part of our manuscript was revised as follows:
From Figure 11, when the processing temperatures increased, E increased first and then decreased gradually. The modulus of elasticity can be regarded as an index to measure the difficulty of elastic deformation of the material. The larger the value, the greater the stress that causes the material to undergo a certain elastic deformation. The greater the material stiffness, the elasticity occurs under certain stress, the smaller the elastic deformation. Based on this, when the processing temperature is 850°C, the E reaches its maximum value in this paper. This result is also mentioned in Ref. 26. It means that the stiffness is the largest and the elastic deformation is the smallest in the design processing temperature [26].
Point 11: “It may be due to cracks occurred in the hardened layer and the hardened layer peeled off at this processing temperature” ok here you make a new speculation but you don’t mention what past studies found, is it similar to yours or different, you need to discuss in more depth your results and findings.
Response 11: Thank the reviewer for the comments. Your suggestions were adopted and the relevant part of our manuscript were revised in manuscript.
The indentation results reveal that both the elastic modulus and the hardness in-crease with the decreasing residual stress of the samples. When the E was large, the amount of elastic recovery was large during the per unit time. It was also indicated that the sample has better resistance to plastic deformation at 850°C. It has resulted that the indentation modulus reached the maximum value at this processing temperature. From Figure 11, when the processing temperature at 950°C, the curve was not smooth and a polyline appeared. It is due to cracks occurred in the hardened layer and the hardened layer peeled off at this processing temperature. This result can be observed in Figure 7 (e).
Point 12: “When the processing temperature is 750°C, the white substance does not exist on the surface.” This claim needs a reference and more explanation, why this happened?
Response 12: Thank the reviewer for the comments. Your suggestions were adopted and the relevant part of our manuscript were revised as follows:
It can be seen from Figure 12 that in addition to carbon fibers, there are also white substances on the surface of some samples. As shown in Figure 12, when the processing temperature is 750°C, the white substance does not exist on the surface. From the sub-sequent experimental results shown in Figure 13, it can be seen that the white sub-stance is caused by the phenomenon of whisker growth. The driving force of the whisker growth phenomenon is the existence of compressive stress [27-29]. When the processing temperature is 750°C, there is tensile stress. At this processing temperature, whisker growth does not occur, and there is no white substance. The specific discussion can be seen from the following discussion.
Point 13: “which may be related to the hardened layer formed.” Again so many speculations, please explain further and support with references, also is this the only possible reason or there might be other ones responsible for the hardened layer?
Response 13: Thank the reviewer for the comments. Your suggestions were adopted and the relevant part of our manuscript were revised as follows:
And when the processing temperature is 750°C, the hardened layer is very thin, the micro-particles will be generated to increase wear when worn. This consideration is supported by the fact that many small particles on the edge of the wear scar were observed in the observation result of wear scars in Figure 15. When the processing temperature at 850°C, the worn surfaces show very limited deformation and smearing, and the wear depth decreased significantly, which is related to the hardened layer formed. The hardened layer plays a role in protecting the substrate Ti. As can be seen from Figure 7, the hardened layer at this processing temperature is the thickest. Comparing with Figure 7, it can be clearly seen that the wear depth was shallower when the hardened layer was thicker. This extraordinary improvement in wear resistance of the treated sample is attributed to the hard surface layer formed during processing (Figure 7).
Point 14: “wear became serious, and obvious” this is a very vague sentence! Define very serious here, it is better to quantify any findings like wear to give clear indication of what is meant by serious here compared to something not serious
Response 14: Thank the reviewer for the comments. Your suggestions were adopted and the relevant part of our manuscript were revised as follows:
Specifically, when the processing temperature at 900°C, the wear width and wear depth is reached to 793 μm and 16.0 μm respectively. In addition, the wear width was wider and the obvious adhesion on the surface of the wear scar. Moreover, it can be seen from the results in Table 5, when the processing temperature increased to 950°C, the width of the wear scar becomes wider and the depth of the wear scar becomes deeper. The maximum values 1024 μm and 19.4 μm are reached respectively. This indicates that the wear has become severe on the surface of the hardened layer. It is a result that the hardened layer peeled off and generates many particles on the surface lead to abrasive wear become serious.
Point 15: “it played a lubricating effect and reduced wear.” Again another speculation but not further explanation or comparing with past studies..
Response 15: Thank the reviewer for the comments. Your suggestions were adopted and the relevant part of our manuscript were revised.
When the processing temperature at 850°C, the worn surfaces show very limited de-formation and smearing, and the wear depth decreased significantly, which is related to the hardened layer formed. The hardened layer plays a role in protecting the substrate Ti. As can be seen from Figure 7, the hardened layer at this processing temperature is the thickest. Comparing with Figure 7, it can be clearly seen that the wear depth was shallower when the hardened layer was thicker. This extraordinary improvement in wear resistance of the treated sample is attributed to the hard surface layer formed during processing (Figure 7). In addition, it can be seen from the 2D profilometric view at this processing temperature that the surface is uneven. This is due to a large amount of carbon fiber exist on the surface. As pointed out above, a surface layer with ultra-high hardness was achieved on the top-most layer with the substrate having gradually decreasing hardness. Such surface formation is beneficial, especially for wear resistance material. Because the hard surface layer increases both adhesive and abrasive wear resistance of soft Ti substrate. The beneath layer just below the top-most layer with relatively low hardness may have a good damping effect which prevents delamination during rubbing, especially under heavy loading conditions. When the processing temperature was higher than 850°C, the wear becomes serious. Specifically, when the processing temperature at 900°C, the wear width and wear depth is reached to 793 μm and 16.0 μm respectively. In addition, the wear width was wider and the obvious adhesion on the surface of the wear scar. Moreover, it can be seen from the results in Table 5, when the processing temperature increased to 950°C, the width of the wear scar becomes wider and the depth of the wear scar becomes deeper. The maximum values 1024 μm and 19.4 μm are reached respectively. This indicates that the wear has become severe on the surface of the hardened layer. It is a result that the hardened layer peeled off and generates many particles on the surface lead to abrasive wear become serious.
Point 16: “it played a lubricating effect and reduced wear.” By how much? It is not clear to me the extent of its effect on the lubrication phenomena.
Response 16: Thank the reviewer for the comments. Your suggestions were adopted and the relevant part of our manuscript were revised.
In addition, Zhou Z. et al. [30] also found that whiskers can play a role in reducing wear. Whiskers have a fibrous structure. In the wear testing, it played a lubricating effect and reduced wear. Zeng Y. et al. also researched that the Mg2B2O5 nanowires that produce the phenomenon of whisker growth can be used as excellent anti-wear additives as the load increases and the wear resistance improves [32]. This is also one of the reasons why the wear width and the wear depth reach the minimum values 385 μm and 2.01 μm respectively, which makes the wear resistance at the processing temperature is 850°C reach the optimal during the designed processing temperature.
Point 17: The results are merely described and is limited to comparing the experimental observation. The authors are encouraged to include a detailed discussion which critically discuss the observations from this investigation with existing literature.
Response 17: Thank the reviewer for the comments. Your suggestions were adopted and the relevant parts were revised.
- Oxidation at all designed temperatures resulted in the formation of a hardened layer, and as the processing temperature increases, the thickness of the hardened layer increased and decreased. The relative thickness depended greatly on the processing temperature, with reaches a maximum thickness at a processing temperature of 850°C.
- Severe plastic deformation with heavy smearing and scratches occurred on the surface of untreated samples during wear testing. The worn surfaces of the oxide samples showed minor deformation and smearing effects at high processing temperature, with minimum wear depth and width for a processing temperature of 850°C.
- This processing reduced the fricative value and coefficient of friction. As the processing temperature increased, the fricative value of the samples decreased initially and then increased. when the processing temperature is 850°C, the fricative value and coefficient of friction reached a minimum value respectively.
- The process used in the present study substantially increased the wear resistance of Ti due to the high surface hardness. The reason why they wear resistance is improved by processing is not only the presence of oxides but also the presence of carbon fibers. Besides, the phenomenon of whisker growth also plays a role in reducing wear.

Reviewer 2 Report
The manuscript entitled: Influence of oxidation processing temperature on the structure, mechanical and tribological properties of titanium using carbon sheet deals with the microstructure and inturn the mechanical and tribological properties of surface processed Ti. I have the following concerns with the manuscript.
- The authors have claimed of using C sheets. However, from Fig. 1 it seems to the C tubes/rods. Can you please clarify?
- The motivation of the present study is not clear nor it is properly highlighted.
- Why does the hardness increase initially with temperature and then show a decrease after 850 C? As if just the peeling effect of you need to consider other effects (since the beta transus temperature of Ti around this temperature)
- Scale bars for all the images in Fig. 8, 10, 12, and 14 should be introduced.
- I do not see all the markings in Fig. 14 are proper. Please double-check!
Author Response
Response to Reviewer 2 Comments
Point 1: The authors have claimed of using C sheets. However, from Fig. 1 it seems to the C tubes/rods. Can you please clarify?
Response 1: Thank the reviewer for the comments. At the macro level, the carbon used is a sheet-like substance, while at the micro level it is fibrous.
Point 2: The motivation of the present study is not clear nor it is properly highlighted.
Response 2: Thank the reviewer for the comments. Your suggestions were adopted and the relevant parts were revised.
Bansal R. et al. reported the study investigates the influence of oxidation temperature for pure titanium to achieve improve corrosion resistance [10]. Maytorena Sánchez A. et al. discussed the formation TiO2 coating can improve the hardness by oxidation [11]. Du H.L. et al. reported the high-temperature corrosion of Ti and Ti-6Al-4V [12]. It found that adding Al into the simple boronized coating is beneficial for the high-temperature oxidation resistance. Most of today's oxidation articles are about the oxidation of alloys. Articles about the oxidation of pure titanium are only a single study of its corrosion or mechanical properties, and there are few studies on the overall structure, mechanical, and tribological properties of pure titanium. In addition, based on Ref. 12, this paper proposes a conjecture about whether the use of carbon cloth will improve the performance of pure titanium during oxidation treatment, and conducts specific studies on the structure, mechanics, and friction and wear properties. Many researchers have research pure titanium preliminarily, but there are few reports about oxidation by using carbon sheet.
Point 3: Why does the hardness increase initially with temperature and then show a decrease after 850°C? As if just the peeling effect of you need to consider other effects (since the beta transus temperature of Ti around this temperature)
Response 2: Thank the reviewer for the comments. Your suggestions were adopted and the relevant part of our manuscript were revised.
However, when the processing temperature increased to 950°C, the H was greatly de-creased due to peeling off of the hardened layer. This was consistent with the SEM observation results. In addition, when the processing temperature is higher than the phase transition temperature, as the processing temperature increases, the movement of molecules becomes more violent, and the diffusion speed of oxygen in the β phase slows down, making the space between titanium atoms and oxygen atoms increase. When the temperature decreases to room temperature, the performance of the sample at this processing temperature is more unstable than the performance of the sample processed under the phase transition temperature, resulting in a decrease in hardness. According to the measured H, the specific indentation depth is shown in Table 2. From Table 2, the processing temperature at 850°C, the indentation depth got the minimum value. And the indentation depth at this time is smaller than the thickness of the hardened layer. From the Table 2, it can be seen that when the processing temperature is 750°C, the indentation depth is the largest, and this indentation depth is greater than the thickness of the hardened layer, so the measured H is greatly affected by the Ti substrate, resulting in the surface at this time got to the low hardness. When the processing temperature is 950°C, although the indentation depth is very large, it can be seen from the results of Figure 7 that the hardened layer peels off. This is the reason for the low the H at this processing temperature.
Describe later, another reason for the higher hardness at 850°C is that a large number of whiskers are produced at this processing temperature. The whiskers have higher hardness, resulting in higher the H at this processing temperature.
Point 4: Scale bars for all the images in Fig. 8, 10, 12, and 14 should be introduced.
Response 4: Thank the reviewer for the comments. Your suggestions were adopted and the relevant part of our manuscript were revised.
Point 5: I do not see all the markings in Fig. 14 are proper. Please double-check!
Response 5: Thank the reviewer for the comments. Your suggestions were adopted and the relevant part of our manuscript was revised.

Reviewer 3 Report
The reviewer comments of the paper «Influence of oxidation processing temperature on the structure, mechanical and tribological properties of titanium using carbon sheet»
- Reviewer
The authors presented an article «Influence of oxidation processing temperature on the structure, mechanical and tribological properties of titanium using carbon sheet». However, there are several points in the article that require further explanation.
Comment 1:
- Experimental
Please describe the details of the experiments. List the chemical composition of Ti in the table. Are there any impurities? What kind? Provide photos or design diagrams of test stands (for mechanical and tribological properties) with a description. How many repetitions of the tests performed? What is the hardness of the material? What about what? What is the coefficient of friction, etc? More details needed.
Comment 2:
- Results and discussion
The quality and resolution of all figures needs to be improved. Redraw the figures 6, 9a, 13, 16 in color.
In Figure 10, c is signed three times. The authors correct everything carefully. Add gaps between all the figures.
From figure 12 it is not clear how a and b differ?
From figure 13 it is not clear where a, b, etc. It is not shown.
Arrange all the figures in the same style font. These differences are especially visible in Figure 9 a and b.
Comment 3:
It will be useful to add a section of Nomenclature in which to sign all the physical quantities and abbreviations encountered in the article. There are many physical quantities in the text and such a section will help to find the description of the necessary element.
For example,
E : Young modulus (GPa)
SEM : Scanning electron microscope
etc.
Comment 4:
The article must be proofread by a native English speaker.
The article is interesting. Authors should carefully study the comments and make improvements to the article step by step. After major changes can an article be considered for publication in the "Metals".
Author Response
Response to Reviewer 3 Comments
Point 1: Please describe the details of the experiments. List the chemical composition of Ti in the table. Are there any impurities? What kind? Provide photos or design diagrams of test stands (for mechanical and tribological properties) with a description. How many repetitions of the tests performed? What is the hardness of the material? What about what? What is the coefficient of friction, etc? More details needed.
Response 1: Thank the reviewer for the comments. Your suggestions were adopted and the relevant part of our manuscript were revised as follows:
The samples used in this study comprised a 50 × 20 × 5 mm3 plate cut from the as-received plate using fine cut machining (HEIWA TECHNICA, HS-45A2). Pure grade-2 Ti (composition in wt.%: 0.2 O, 0.003 N, 0.013 H, 0.25 Fe, 0.008 C, and remainder Ti) was used as the substrate material. The surface to be treated of the Ti was finished by grinding with emery paper (grade 4000). The pure titanium is heated by the air furnace in the atmosphere by using the carbon sheet. The schematic diagram of the sample setup was shown in Figure 1. The scanning electron microscope (SEM, HITA-CHI, S-3000N) image of the carbon sheet was shown in Figure 2. The titanium plate was placed between two pieces of carbon sheet and the load at 1 MPa was applied to it. Processing temperatures were changed from 750°C to 950°C for 2 h in the atmosphere. To determine the compound formed on the surface after processing, the treated sur-faces were investigated using an X-ray diffractometer (XRD, Rigaku, 2200VF) with CuKα radiation working at the optimum voltage of 32 kV and anodic current of 20 mA. Furthermore, the cross-section of treated samples was prepared by ion milling (HI-TACHI, IM4000) with 7200 s. Microstructural and morphological characteristics of hardened layers were examined using a SEM. Surface hardness testing was carried out throughout the hardened layers of the samples using a Vicker hardness tester (SHI-MADZU, HMV-1) with the applied load is 0.98 N to gain the hardness of the hardened layer more accurately and at a dwell time of 15 s. In order to reduce the measurement error, the sample was measured for 5 times and averaged. Ball-on-disk dry sliding tests were performed at room temperature to evaluate the tribological properties of the samples using a tribometer (Rhesca, FPR-2000) with a zirconium dioxide (ZrO2) ball and a counter-face with a radius of 2380 μm. Samples with the dimensions of 50 × 20 × 5 mm3 were used for wear testing with a sliding linear speed of 200 mm/s, an applied force (loading force during the wear testing) of 4.9 N, and a test time of 3600 s (corresponding to a sliding distance of 720 m). The experimental apparatus of the wear and fricative testing was shown in Figure 3. All tests were performed at room temperature. The microstructural and morphological features of the wear track after wear testing were examined using SEM under secondary electron imaging mode at a voltage of 15 kV. The fricative value is measured by a condenser microphone (TENMARS, TM-103) installed 20 mm above the contact point between the ball and sample. The depth pro-file of the sliding portion after the wear testing was measured by a laser microscope. In order to reduce the measurement error, the sample was measured for 4 times and averaged.
Figure 1. Schematic diagram of the sample setup.
Figure 2. SEM image of carbon sheet.
Figure 3. Experimental apparatus of the wear and fricative testing.
Point 2: The quality and resolution of all figures needs to be improved. Redraw the figures 6, 9a, 13, 16 in color.
Response 2: Thank the reviewer for the comments. Your suggestions were adopted and the relevant part of our manuscript were revised.
Point 3: In Figure 10, c is signed three times. The authors correct everything carefully. Add gaps between all the figures.
From figure 12 it is not clear how a and b differ?
From figure 13 it is not clear where a, b, etc. It is not shown.
Arrange all the figures in the same style font. These differences are especially visible in Figure 9 a and b.
Response 3: Thank the reviewer for the comments. Your suggestions were adopted and the relevant part of our manuscript were revised.
Point 4: It will be useful to add a section of Nomenclature in which to sign all the physical quantities and abbreviations encountered in the article. There are many physical quantities in the text and such a section will help to find the description of the necessary element.
For example,
E: Young modulus (GPa)
SEM: Scanning electron microscope
etc.
Response 4: Thank the reviewer for the comments. Your suggestions were adopted and the relevant part of our manuscript was revised.
Point 5: The article must be proofread by a native English speaker.
Response 5: Thank the reviewer for the comments. Your suggestions were adopted and the relevant part of our manuscript was revised.

Round 2
Reviewer 1 Report
Combine figures 1-3 in one figure
figure 6 quality is low and must be improved
figure 14 add some arrwos and text to tell the readers what are they looking at there
reduce the size of figure 15
Author Response
Response to Reviewer 1 Comments
Point 1: Combine figures 1-3 in one figure
Response 1: Thank the reviewer for the comments. Your suggestions were adopted and the relevant part of our manuscript was revised as follows:
(a) (b)
(c)
Figure 1. Details of the experimental work (a) schematic diagram of the sample setup, (b) experimental apparatus of the wear and fricative testing and (c) SEM carbon sheet image.
Point 2: figure 6 quality is low and must be improved
Response 2: Thank the reviewer for the comments. Your suggestions were adopted and the relevant part of our manuscript was revised as follows:
This figure is quoted from the reference. The quality of this figure is lower than that of the figures made by yourself.
Figure 4. Titanium-carbon phase diagram. [9].
Point 3: figure 14 add some arrwos and text to tell the readers what are they looking at there
Response 3: Thank the reviewer for the comments. Your suggestions were adopted and the relevant part of our manuscript were revised as follows:
(a) (b)
Figure 12. SEM micrographs of the carbon fiber cross-section with the processing temperature at 850°C with different locations (a) and (b).
Point 4: reduce the size of figure 15
Response 4: Thank the reviewer for the comments. Your suggestions were adopted and the relevant part of our manuscript was revised in the manuscript.

Reviewer 2 Report
The authors have satisfactorily addressed most of the comments raised and I may recommend the manuscript for publication in the present form.
Author Response
Thank the reviewer for the comments.
Reviewer 3 Report
The authors have improved the article according to the comments. The article can now be published.
Author Response
Thank the reviewer for the comments.